# FEDERATED LEARNING UNDER LABEL SHIFTS WITH GUARANTEES

## ABSTRACT

We consider the problem of training a global model in a distributed setting and develop an unbiased estimate of the overall *true risk* minimizer of multiple clients under challenging inter-client and intra-client *label shifts* as a stepping stone to provably address distribution shifts in real world. We generalize the family of Maximum Likelihood Label Shift (MLLS) density estimation methods inspired by a board family of Integral Probability Metrics and introduce the Variational Regularized Label Shift (VRLS) family of density ratio estimation methods and show all MLLS methods are special cases of VRLS under specific latent spaces. Our theory shows high-probability estimation error bounds achieved through a versatile regularization term in VRLS. Our extensive numerical experiments demonstrate that VRLS establishes *a new SotA in density ratio estimation* surpassing all baselines in MNIST, Fashion MNIST, CIFAR-10 datasets and *relaxed label shifts* as a proxy of real-world settings. In distributed settings, our importance-weighted empirical risk minimization with VRLS outperforms federated averaging and other baselines in imbalanced settings under drastic and challenging label shifts.[1]

## 1  INTRODUCTION

Supervised learning trains a machine learning model, e.g., a neural network, given access to training samples, each as a pair of (feature, label). The cornerstone of classical learning theory hinges on the assumption that data samples, both in training and test time, are independently and identically distributed (i.i.d.). However, such i.i.d. premise becomes overly idealistic in the context of real-world settings where training and test data can be drawn from different distributions, which may change dynamically as the operation environment evolves. As initial attempts to address joint distribution shifts in real world, two models for distribution shifts are common: covariate shift and label

shift (Sugiyama et al., 2007; Lipton et al., 2018; Garg et al., 2022; Mani et al., 2022; Zhou et al., 2023). Covariate shift assumes the conditional distribution of labels $\mathbf{y}$ given features $\mathbf{x}$ remains the same across train and test, i.e., $p^{\mathrm{tr}}(\mathbf{y}|\mathbf{x}) = p^{\mathrm{te}}(\mathbf{y}|\mathbf{x}) := p(\mathbf{y}|\mathbf{x})$. Label shift assumes the marginal train distribution $p^{\mathrm{tr}}(\mathbf{y})$ and test distribution $p^{\mathrm{te}}(\mathbf{y})$ can be arbitrarily different; while the conditional distribution $p(\mathbf{x}|\mathbf{y})$ remains relatively stable. The evolving COVID-19 pandemic, characterized by fluctuating infection rates across regions, serves as a tangible example of label shift. This scenario can aptly be modeled under a federated learning framework, given the localized nature of outbreaks and developments. Another example for label shift is variations in text sentiments over time with economy status. Intuitively, label shift arises when labels $\mathbf{y}$

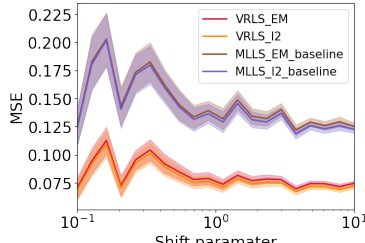

**Figure 1:** *MSE on CIFAR-10 under Relaxed Label Shift.*

cause features $\mathbf{x}$ (Zhang et al., 2013; Lipton et al., 2018; Rabanser et al., 2019; Garg et al., 2020; 2023). In this paper, we focus on addressing label shift both in a single client and federated settings.

Federated Learning (FL) emerges as a robust and privacy-preserving framework for collaboratively learning a machine learning model across multiple clients such as hospitals and smartphones without sharing clients' local data, where distribution shifts generally exist. To train the model, current FL

---

[1]We will release the code publicly along with the final version.

literature has primarily adopted the empirical risk minimization (ERM), tacitly assuming identity of training and test data distributions for each client. This simplifying assumption does not address statistical heterogeneity in realistic settings. This heterogeneity is manifested in the form of inter-client distribution shifts (variations among clients) and intra-client distribution shifts (variations within a single client). To rigorously and properly address statistical heterogeneity in realistic FL settings, it is crucial to look beyond ERM. In this paper, we propose an Importance-weighting ERM (IW-ERM) framework to address joint intra-client and inter-client label shifts with *generalization guarantees*. Our IW-ERM handles intra-processor and inter-processor distribution shifts in any *distributed platform* including cloud computing, supercomputing, and volunteer computing.

The main technical challenge to provably handle label shifts is to efficiently and accurately estimate density ratios $p^{\text{te}}(\mathbf{y})/p^{\text{tr}}(\mathbf{y})$ for all labels taking into account both intra- and inter-client label shifts. Having access to labeled training data and sufficient number of unlabeled test samples, techniques such as Black Box Shift Estimation (BBSE) of Saerens et al. (2002); Lipton et al. (2018); Rabanser et al. (2019) and Regularized Learning under Label Shift (RLLS) of Azizzadenesheli et al. (2019) employ a confusion matrix strategy to estimate these ratios. Garg et al. (2020) has proposed the Maximum Likelihood Label Shift Estimation (MLLS) method, which unifies both BBSE and RLLS with a maximum likelihood estimation framework and learning a predefined predictor $f$ through either expectation-maximization (EM) of Saerens et al. (2002) or other first-order optimization variants. Empirical results showcase that MLLS paired with a post-hoc calibration method, e.g, Bias-Corrected Calibration (BCT) of Alexandari et al. (2020) outperforms vanilla BBSE.

A canonical calibrated predictor is essential for ratio estimation in MLLS (Garg et al., 2020). Concurrently, domain alignment stands as a common strategy in handling domain adaptation issues (Fernando et al., 2013; Kumar et al., 2018). Nonetheless, applying alignment directly in FL settings presents challenges; the variability of label shifts across clients and accessing unlabeled test data violates stringent privacy requirements. To improve estimation error and generalization of the MLLS family, we introduce a distinct latent variable $\mathbf{z}$, different from that in MLLS family and reformulate the unified MLLS by incorporating a regularization term for the predictor $f$. This term ensures consistency of predicted labels in our designated latent space $\mathbf{z}$ between two predefined input latent distributions. Subsequently, we employ parameterized variational distributions to approximate the true input latent distributions to minimize the discrepancy within the latent space $\mathbf{z}$ under the worst-case scenario. The MLLS family can be degenerated from our method under special settings.

In FL, we propose an Importance Weighted ERM (IW-ERM) framework with generalization guarantees and same privacy guarantees as ERM baselines. Our method of density ratio estimation trains the pre-defined predictor *exclusively using local training data*. For ratio estimation, the communication between clients involves only the estimated marginal label distribution, instead of data, ensuring privacy preservation and negligible communication overhead.

## 1.1 SUMMARY OF CONTRIBUTIONS

- We show that performance disparity among variants of MLLS reported in (Garg et al., 2020) is indeed due to implementation issues. We empirically demonstrate that MLLS, under varying shift parameters, when paired with both EM and convex optimization, outperforms the gradient-based methods and reach to the same optimal solution, which matches the theory (Fig. 1).

- We propose a novel method for density ratio estimation in two stages that employs variational inference with dual latent variables. This method uniquely combines a Dirichlet prior and integrates a versatile regularization term for the predictor, rooted in the broad family of Integral Probability Metrics (IPM) (Sriperumbudur et al., 2009) and metric space similarity comparisons. Moreover, we implement our method using Maximum Mean Discrepancy (MMD) (Gretton et al., 2012) with various kernels.

- We show importance weighting does not negatively impact the convergence rates and communication guarantees in a broad range of importance optimization settings.

- Our empirical results over MNIST, Fashion MNIST, and CIFAR-10 datasets showcase the superior performance of our ratio estimation in reducing estimation errors compared all MLLS baselines. Furthermore, our approach exhibits enhanced robustness in scenarios involving *relaxed label shifts* (Fig. 1) and demonstrates improved performance under stringent worst-case sampling conditions.

- We have successfully incorporated our ratio estimation into FL, demonstrating close performance to an upper bound with true ratios on Fashion MNIST and CIFAR-10 datasets with 5 and 100 clients. Our IW-ERM framework adeptly handles both inter- and intra-client label shifts while preserving privacy and ensuring generalization guarantees and shows up to 20% improvements in terms of average test error over current baselines.

## 1.2 RELATED WORK

In this section, we overview summary of related work. See Appendix A for complete discussion.

**Federated Learning.** The current FL research predominantly centers around the minimization of the empirical risk, operating under the assumption that each client maintains the same training/test data distribution and heuristic-based personalization methods to handle statistical heterogeneity across clients (Kairouz et al., 2021). In contrast, we aim to minimize overall test error under intra-client and inter-client label shifts, which is a major challenge in real-world scenarios (Garg et al., 2023).

**Importance Weighting, Label Shift, and MLLS Family.** Shimodaira (2000) has shown that the IW-ERM estimator is asymptotically unbiased. Recently, Ramezani-Kebrya et al. (2023) introduced FTW-ERM, which integrates density ratio estimation to handle covariate shifts in FL. In this paper, we focus on label shifts and show that our IW-ERM with VRLS performs very close to the upper bound under true ratios in (Ramezani-Kebrya et al., 2023) without sharing any local data.

Density ratio estimation for label shifts has been tackled by solving a linear system (Lipton et al., 2018; Azizzadenesheli et al., 2019) and by minimizing distribution divergence (Garg et al., 2020) for one client. In (Saerens et al., 2002; Lipton et al., 2018; Azizzadenesheli et al., 2019; Garg et al., 2020), the conditional distribution $p(\mathbf{x}|\mathbf{y})$ is strictly constant across train and test. Saerens et al. (2002); Lipton et al. (2018) have proposed BBSE and RLLS by designating a discrete latent space $\mathbf{z}$ and introduce a confusion matrix-based estimation method to compute the ratios. BBSE is straightforward and has been proven consistency even when the predictor is not calibrated. However, its subpar performance is attributed to the information loss inherent in the confusion matrix (Garg et al., 2020). Garg et al. (2020) has introduced MLLS with a continuous latent space, resulting in a significant enhancement in estimation performance, especially when combined with a post-hoc calibration method (Shrikumar et al., 2019). In this work, we focus on overall generalization performance on multiple clients under both intra-client and inter-client label shifts in FL compared to (Garg et al., 2020), we consider a more general latent space and extend our consideration to relaxed label shift.

**Notation:** We use $\mathbb{E}[\cdot]$ to denote the expectation and $\|\cdot\|$ to represent the Euclidean norm of a vector. We use lower-case bold font to denote vectors. Sets and scalars are represented by calligraphic and standard fonts, respectively. We use $[n]$ to denote $\{1, \ldots, n\}$ for an integer $n$. We use $\lesssim$ to ignore terms up to constants and logarithmic factors.

## 2 LABEL SHIFT AND IW-ERM

Let $\mathcal{X} \subseteq \mathbb{R}^{d_0}$ be a compact metric space for input features, $\mathcal{Y}$ be a discrete label space with $|\mathcal{Y}| = m$, and $K$ be the number of clients in an FL setting. Let $\mathcal{S}_k = \{(x_{k,i}^{\mathrm{tr}}, y_{k,i}^{\mathrm{tr}})\}_{i=1}^{n_k^{\mathrm{tr}}}$ denote the training set of client $k$ with $n_k^{\mathrm{tr}}$ samples drawn i.i.d from a probability distribution $p_k^{\mathrm{tr}}$ on $\mathcal{X} \times \mathcal{Y}$. The test data of client $k$, is drawn from another probability distribution $p_k^{\mathrm{te}}$ on $\mathcal{X} \times \mathcal{Y}$. We assume that the class-conditional distribution $p_k^{\mathrm{tr}}(\mathbf{x}|\mathbf{y}) = p_k^{\mathrm{te}}(\mathbf{x}|\mathbf{y}) := p(\mathbf{x}|\mathbf{y})$ remains the same for all client $k$. This is a common assumption and holds when label shifts primarily affect the prior distribution of the labels $p(\mathbf{y})$ rather than the underlying feature distribution given the labels and holds when the way features are generated given a label remains constant (Zadrozny, 2004; Huang et al., 2006; Sugiyama et al., 2007). Note that $p_k^{\mathrm{tr}}(\mathbf{y})$ and $p_k^{\mathrm{te}}(\mathbf{y})$ can be arbitrarily different, which gives rise to intra-client and inter-client *label shifts* (Zadrozny, 2004; Huang et al., 2006; Sugiyama et al., 2007; Garg et al., 2023). To better model realistic settings, in Section 5, we extend our consideration to relaxed label shift notion of Garg et al. (2023).

Our primary goal is to find an unbiased estimate of the overall *true risk* minimizer of multiple clients under intra-client and inter-client *label shifts*, i.e., to find a hypothesis $h_{\mathbf{w}} \in \mathcal{H} : \mathcal{X} \to \mathcal{Y}$, e.g., a neural network parameterized by $\mathbf{w}$, such that $h_{\mathbf{w}}(\mathbf{x})$ is a good approximation of the label $\mathbf{y} \in \mathcal{Y}$

corresponding to a fresh sample $\mathbf{x} \in \mathcal{X}$ drawn from the aggregated *test* data. Let $\ell : \mathcal{X} \times \mathcal{Y} \to \mathbb{R}_+$ denote a loss function. Client $k$ aims to learn a hypothesis $h_{\mathbf{w}}$ that minimizes its true (expected) risk:

$$R_k(h_{\mathbf{w}}) = \mathbb{E}_{(\mathbf{x},\mathbf{y}) \sim p_k^{\mathrm{te}}(\mathbf{x},\mathbf{y})}[\ell(h_{\mathbf{w}}(\mathbf{x}), \mathbf{y})].$$

We now modify the classical ERM and formulate IW-ERM to find a predictor that minimizes the aggregate true risk over all clients:

$$\min_{h_{\mathbf{w}} \in \mathcal{H}} \sum_{k=1}^{K} \frac{\lambda_k}{n_k^{\mathrm{tr}}} \sum_{i=1}^{n_k^{\mathrm{tr}}} \frac{\sum_{j=1}^{K} p_j^{\mathrm{te}}(\mathbf{y}_{k,i}^{\mathrm{tr}})}{p_k^{\mathrm{tr}}(\mathbf{y}_{k,i}^{\mathrm{tr}})} \ell(h_{\mathbf{w}}(\mathbf{x}_{k,i}^{\mathrm{tr}}), \mathbf{y}_{k,i}^{\mathrm{tr}}) \qquad \text{(IW-ERM)}$$

where $\sum_{k=1}^{K} \lambda_k = 1$ and $\lambda_k \geq 0$.

**Proposition 1.** *Let $\boldsymbol{\lambda} \succeq \mathbf{0}$ with $\mathbf{1}^\top \boldsymbol{\lambda} = 1$. The* (IW-ERM) *is consistent and the learned function $h_{\mathbf{w}}$ converges in probability to the optimal function in terms of minimizing the overall true risk $\sum_{k=1}^{K} R_k$.*

*Proof.* See Appendix C. The convergence in probability is followed the arguments in (Shimodaira, 2000)[Section 3] and (Sugiyama et al., 2007)[Section 2.2] using the law of large numbers. ∎

## 3 DENSITY RATIO ESTIMATION

To solve Eq. (IW-ERM), client $k$ should have access to an accurate estimate of $r_k(\mathbf{y}) = \sum_{j=1}^{K} p_j^{\mathrm{te}}(\mathbf{y})/p_k^{\mathrm{tr}}(\mathbf{y})$. For rotational simplicity, we first consider the scenario where we have only one client under label shifts and then extend to multiple clients. The goal is to estimate this ratio:

$$r(\mathbf{y}) = \frac{p^{\mathrm{te}}(\mathbf{y})}{p^{\mathrm{tr}}(\mathbf{y})}. \qquad \text{(Ratio)}$$

### 3.1 REGULARIZED RATIO ESTIMATION

We introduce a distinct latent space, denoted as $\mathcal{Z}$, which represents the kernel-transformed predictor output. Subsequently, we define a novel pre-determined function $\mathbf{h}(\mathbf{x}) = \mathbf{g}(\mathbf{f}(\mathbf{x}))$ to approximate $p^{\mathrm{tr}}(\mathbf{z}|\mathbf{x}) = \sum_y p^{\mathrm{tr}}(\mathbf{z}|\mathbf{y})p^{\mathrm{tr}}(\mathbf{y}|\mathbf{x})$ as an alternative to using $f(\mathbf{x})$ for approximating $p^{\mathrm{tr}}(\mathbf{y}|\mathbf{x})$ as in BBSE and MLLS. Within this framework, the predictor $f$ is employed to approximate $p^{\mathrm{tr}}(\mathbf{y}|\mathbf{x})$, and the outer mapping $g$ is used to approximate $p^{\mathrm{tr}}(\mathbf{z}|\mathbf{y})$. Given these approximations, we formulate the following optimization problem to estimate density ratios:

$$\mathbf{r}_f := \arg\max_{\mathbf{r} \in \mathcal{R}} \left( \mathbb{E}_{\mathrm{te}} \left[ \log(f(\mathbf{x})^T \mathbf{r}) + \log g(f(\mathbf{x})) \right] \right) \qquad \text{(Reg-Est)}$$

where $\mathbf{r}$ includes estimates of Eq. (Ratio) for all classes and $\mathcal{R}$ is the constraint set defined similar to (Garg et al., 2020). In Eq. (Reg-Est), the first term aligns with the MLLS objective, while the second term serves as a regularization component for the predictor $f$. This regularization is defined over the test distribution within the latent space $\mathcal{Z}$. Given a fixed predictor $f$ w.r.t. $\mathbf{r}$, Eq. (Reg-Est) degenerates to MLLS family objective. Recognizing the inherent regularization properties, we strategically incorporate the regularization term into the training process *without accessing to test distribution*. This allows us to utilize it as a robust regularization component, thereby improving estimation error. By replacing $g$ with $\delta_{\arg\max(\cdot)}$, Eq. (Reg-Est) reduces to MLLS_CM. Removing the regularization term, e.g., by assuming $g(f(\mathbf{x}))$ is deterministic, Eq. (Reg-Est) yields MLLS family of Eq. (B.6) in Appendix B.

To solve Eq. (Reg-Est), we first focus on the second term and optimize the predictor $f$ using training samples with the cross-entropy loss and a regularization parameter $\lambda > 0$ in Eq. (3.1). Upon obtaining improved $f$ in Eq. (Reg-Est), the ratios are obtained by Eq. (3.2):

$$\boldsymbol{\theta}^\star = \arg\min_{\boldsymbol{\theta}} \mathbb{E}_{\mathrm{tr}} \left[ \ell_{\mathrm{CE}}(f(\mathbf{x}; \boldsymbol{\theta}), \mathbf{y}) - \lambda \cdot \log g(f(\mathbf{x}; \boldsymbol{\theta})) \right], \qquad (3.1)$$

$$\mathbf{r}_{f^\star} = \arg\max_{\mathbf{r} \in \mathcal{R}} \mathbb{E}_{\mathrm{te}} \left[ \log(f(\mathbf{x}; \boldsymbol{\theta}^\star)^\top \mathbf{r}) \right]. \qquad (3.2)$$

## 3.2 Variational Approximation with Worst-case Sampling

In Eq. (3.1), we establish the following lower bound on the second term by introducing two independent dual latent variables $\mathbf{z}_1$ and $\mathbf{z}_2$:

$$\mathbb{E}_{\mathrm{tr}}[\log g(f(\mathbf{x}))] \geq \mathbb{E}_{\mathrm{tr}}\mathbb{E}_{q_{(\mathbf{z}_1,\mathbf{z}_2|\mathbf{x})}}[\log(g(f(\mathbf{x})))] + \mathrm{KL}\left(q(\mathbf{z}_1,\mathbf{z}_2 \mid \mathbf{x})\|p(\mathbf{z}_1,\mathbf{z}_2 \mid \mathbf{x})\right). \quad (3.3)$$

Given that $g(f(\mathbf{x}))$ is strictly positive with a radial basis function (RBF) kernel selection, $q(\mathbf{z}_1,\mathbf{z}_2|\mathbf{x})$ serves as an approximation to the true $p(\mathbf{z}_1,\mathbf{z}_2|\mathbf{x})$, and $\mathrm{KL}\left(q(\mathbf{z}_1,\mathbf{z}_2|\mathbf{x})\|p(\mathbf{z}_1,\mathbf{z}_2|\mathbf{x})\right)$ is positive. We then optimize the lower bound by aligning $g(f(\mathbf{x}))$ with $\mathbf{z}_1$ and $\mathbf{z}_2$ in $\mathcal{Z}$ and minimizing some loss function $\ell(g(f(\mathbf{x})))$.

In label shift contexts, variations in $p(\mathbf{y})$ influence $p(\mathbf{x})$ through $p(\mathbf{x}) = \sum_{\mathbf{y}} p(\mathbf{x}|\mathbf{y})p(\mathbf{y})$. We adjust to label distribution changes with a latent distribution parameterized by a Dirichlet distribution and controlling parameter $\alpha$ over all classes. We approximate the true posteriors $p(\mathbf{z}_1|\mathbf{x})$ and $p(\mathbf{z}_2|\mathbf{x})$ with variational distributions $q(\mathbf{z}_1|\mathbf{x},\alpha)$ and $q(\mathbf{z}_2|\mathbf{x},\alpha)$. Hence, the objective Eq. (3.1) becomes:

$$\arg\min_{\boldsymbol{\theta}} \mathbb{E}_{\mathrm{tr}}\left[\ell_{\mathrm{CE}}(f(\mathbf{x};\boldsymbol{\theta}),\mathbf{y}) + \lambda \cdot \mathbb{E}_{q(\mathbf{z}_1,\mathbf{z}_2|\mathbf{x},\alpha)}\left[\ell(g(f(\mathbf{x};\boldsymbol{\theta})))\right]\right]. \quad (3.4)$$

### 3.2.1 Worst-case Sampling

To enhance the robustness w.r.t. $q(\mathbf{z}_1,\mathbf{z}_2 \mid \mathbf{x},\alpha)$, we generalize the objective in Eq. (3.4) as follows (Zhu et al., 2021):

$$\boldsymbol{\theta}^{\star} = \arg\min_{\boldsymbol{\theta}} \mathbb{E}_{\mathrm{tr}}\left[\ell_{\mathrm{CE}}(f(\mathbf{x};\boldsymbol{\theta}),\mathbf{y}) + \lambda \cdot \arg\max_{q(\mathbf{z}_1,\mathbf{z}_2|\mathbf{x},\alpha)} \mathbb{E}_{q(\mathbf{z}_1,\mathbf{z}_2|\mathbf{x},\alpha)}\ell(g(f(\mathbf{x};\boldsymbol{\theta})))\right] \quad (3.5)$$

where the worst-case latent variables $\mathbf{z}_1,\mathbf{z}_2$ are identified by maximizing over $\alpha$ while minimizing the loss function $\ell(g(f(\mathbf{x};\boldsymbol{\theta}))$.

Direct optimization of $\alpha$ to pinpoint the worst-case latent distributions is challenging due to the non-differentiable nature of the process involving the determination of label marginal distribution $p(\mathbf{y})$, and data sampling. This challenge is mitigated by setting $\alpha$ based on a narrow normal distribution around zero for "Pseudo-test" samples and $\alpha = \infty$ for "Pseudo-training" samples. Here, under $\lambda = 0$, Eq. (3.5) degenerates to the predictor in MLLS family.

## 3.3 Similarity comparison with IPM

For the loss function $\ell(g(f(\mathbf{x};\boldsymbol{\theta})))$ in Eq. (3.5), we introduce a function $g$ acting as a kernel function, defining similarity within the Reproducing Kernel Hilbert Space (RKHS) (Zhu et al., 2021) using a selected kernel and the MMD loss (Gretton et al., 2012):

$$\begin{aligned}
\mathbb{E}_q\left[\ell(g(f(\mathbf{x};\boldsymbol{\theta})))\right] &= \mathbb{E}_q\left[\mathrm{MMD}_f^2(\mathbf{z}_1,\mathbf{z}_2 \mid \mathbf{x})\right] \\
&:= \mathbb{E}_{q_2}[k(f(\mathbf{x}),f(\mathbf{x}))] - 2\mathbb{E}_q[k(f(\mathbf{x}),f(\mathbf{x}))] + \mathbb{E}_{q_1}[k(f(\mathbf{x}),f(\mathbf{x}))]
\end{aligned} \quad (3.6)$$

where $k(\cdot,\cdot)$ denotes the RKHS defined by the kernel function $g$, and $\mathbb{E}_q$, $\mathbb{E}_{q_1}$, and $\mathbb{E}_{q_2}$ are notations for $\mathbb{E}_{q(\mathbf{z}_1,\mathbf{z}_2|\mathbf{x},\alpha)}$, $\mathbb{E}_{q(\mathbf{z}_1|\mathbf{x},\alpha)}$, and $\mathbb{E}_{q(\mathbf{z}_2|\mathbf{x},\alpha)}$, respectively. Importantly, we compute the Gram matrix using data from both $q_1$ and $q_2$, rather than explicitly computing $g(f(\mathbf{x}))$. The VRLS steps are in Algorithm 1 in Appendix D.

## 3.4 Ratio Estimation for FL under Label Shifts

Having defined our density ratio estimation under label shifts, we further generalize this estimation to FL settings under stringent privacy requirements. To estimate the ratios in Eq. (IW-ERM), each client could potentially first train a predictor on local training data using Eq. (3.4) without sharing any local data. The server then broadcasts the parameters of the predictor to each client. Clients employ their local data to estimate both test and training label marginal distributions using the EM method. This approach avoids the need to share any local data, with only test label marginal distributions being shared among all clients. The ratios in Eq. (IW-ERM) are calculated by clients locally and in parallel. The ratio estimation without sharing local data is described in Listing 1 in Appendix D. Experiments in Section 5 validate that our IW-ERM framework significantly improves average test error without sharing any local data and with negligible communication and computation overhead over the baselines.

## 4 THEORETICAL GUARANTEES

In this section, we provide guarantees on the finite sample errors incurred during the estimation of $\mathbf{r}_{f^\star}$ and $\boldsymbol{\theta}^\star$. Indeed, in practice, we only dispose of a finite number of labeled samples $\{(\mathbf{x}_i, \mathbf{y}_i)\}_{i=1}^n$. These samples serve to compute the following estimates: $\hat{\boldsymbol{\theta}}_n = \underset{\boldsymbol{\theta} \in \Theta}{\arg\min} \frac{1}{n} \sum_{i=1}^n [\ell_{\mathrm{CE}}(f(\mathbf{x}_i; \boldsymbol{\theta}), \mathbf{y}_i) - \lambda \cdot \log g(f(\mathbf{x}_i; \boldsymbol{\theta}))]$, and $\hat{\mathbf{r}}_n = \underset{\mathbf{r} \in \mathcal{R}}{\arg\max} \frac{1}{n} \sum_{i=1}^n \left[ \log(f(\mathbf{x}; \hat{\boldsymbol{\theta}}_n)^\top \mathbf{r}) \right]$. We will show that the errors of these estimates can be controlled. The following assumptions are necessary to establish our results.

**Assumption 1** (Boundedness). The data and the parameter space $\Theta$ are bounded, i.e, there exists $b_\mathcal{X}, b_\Theta > 0$ such that

$$\forall \mathbf{x} \in \mathcal{X}, \ \|\mathbf{x}\|_2 \leq b_\mathcal{X} \qquad \text{and} \qquad \forall \boldsymbol{\theta} \in \Theta, \ \|\boldsymbol{\theta}\|_2 \leq b_\Theta.$$

**Assumption 2** (Calibration). Let $\boldsymbol{\theta}^\star$ be as defined in Eq. (3.5). There exists $\mu > 0$ such that

$$\mathbb{E}\left[ f(\mathbf{x}; \boldsymbol{\theta}^\star) f(\mathbf{x}; \boldsymbol{\theta}^\star)^\top \right] \succeq \mu \mathbf{I}_m.$$

The calibration Assumption 2 first appears in Garg et al. (2020). It is necessary for the ratio estimation procedure to be consistent and we refer the reader to Section 4.3 of Garg et al. (2020) for more details. We further need Assumption 1 because, unlike (Garg et al., 2020), the empirical estimator $\hat{\mathbf{r}}_n$ is estimated using another estimator $\hat{\boldsymbol{\theta}}_n$. Uniform bounds are therefore needed to control finite sample error as we cannot directly apply concentration inequalities, as is done in the proof of (Garg et al., 2020, Lemma 3), since we do not have independence of the terms appearing in the empirical sums. We nonetheless prove a similar result in the following theorem.

**Theorem 1.** *Let $\delta \in (0, 1)$ and $\mathcal{F} := \{\mathbf{x} \mapsto \mathbf{r}^\top f(\mathbf{x}, \boldsymbol{\theta}), \ (\mathbf{r}, \boldsymbol{\theta}) \in \mathcal{R} \times \Theta\}$. Under Assumptions 1 and 2, there exists constants $L > 0, B > 0$ such that with probability at least $1 - \delta$:*

$$\|\hat{\mathbf{r}}_n - \mathbf{r}_{f^\star}\|_2 \leq \frac{2}{\mu p_{\min}} \left( \frac{4}{\sqrt{n}} \mathrm{Rad}(\mathcal{F}) + 4B\sqrt{\frac{\log(4/\delta)}{n}} \right) + \frac{4L}{\mu p_{\min}} \mathbb{E}\left[ \|\hat{\boldsymbol{\theta}}_n - \boldsymbol{\theta}^\star\|_2 \right].$$

*where $p_{\min} = \min_\mathbf{y} p_{\mathrm{tr}}(\mathbf{y})$ and $\mathrm{Rad}(\mathcal{F}) = \frac{1}{\sqrt{n}} \mathbb{E}_{\sigma_1, \dots, \sigma_n} \left[ \sup_{(\mathbf{r}, \boldsymbol{\theta}) \in \mathcal{R} \times \Theta} \left| \sum_{i=1}^n \sigma_i \mathbf{r}^\top f(\mathbf{x}_i, \boldsymbol{\theta}) \right| \right].$*

The Rademacher complexity appearing in the bound will depend on the architecture chosen for $f$. Moreover as regularization often encourages lower complexity functions, this complexity can be reduced because of the presence of the regularization term in the estimation of $\boldsymbol{\theta}$ in our setting.

We now establish convergence rates for Eq. (IW-ERM) with VRLS and show our proposed importance weighting achieves *the same rates* with the data-dependent *constant terms* increase linearly with $\max_{\mathbf{y} \in \mathcal{Y}} \sup_f r_f(\mathbf{y}) = r_{\max}$ under negligible communication overhead over the baseline ERM-solvers without importance weighting. In Appendix F, we establish tight convergence rates and communication guarantees for Eq. (IW-ERM) with VRLS in a broad range of importance optimization settings including convex optimization, second-order differentiability, composite optimization with proximal operator, optimization with adaptive step-sizes, and nonconvex optimization, along the lines of e.g., (Woodworth et al., 2020; Haddadpour et al., 2021; Glasgow et al., 2022; Liu et al., 2023; Hu and Huang, 2023; Wu et al., 2023; Liu et al., 2023).

**Theorem 2** (Convergence-Communication). *Let $\max_{\mathbf{y} \in \mathcal{Y}} \sup_f r_f(\mathbf{y}) = r_{\max}$. Suppose Algorithm 2 is run for $T$ iterations. Then Algorithm 2 achieves a convergence rate of $\mathcal{O}(r_{\max} h(T))$ where $\mathcal{O}(h(T))$ denotes the rate of ERM-solver baseline without importance weighting. Throughout the course of optimization, Algorithm 2 has the same overall communication guarantees as the baseline.*

Theorem 2 shows that importance weighting does not negatively impact the convergence rates and communication guarantees throughout the course of optimization.

## 5 EXPERIMENTS

The experiments are roughly split into two sections: 1) Density ratio estimation performed on a single client under intra-client label shift; 2) FL settings with 5 and 100 clients.

**Density Ratio Estimation.** We begin by evaluating VRLS on MNIST (LeCun et al., 1998) and CIFAR-10 (Krizhevsky) datasets in a single-client scenario, focusing on estimation error. We simulate the test dataset using a Dirichlet distribution with varying alpha parameters, following the setup of (Lipton et al., 2018). Higher $\alpha$ implies a smoother transition, and lower values indicate abrupt shifts. The training dataset is uniformly distributed across all classes.

Initially, with a sample size of 5,000, we investigate 20 $\alpha$ values in the range of $10^{-1}$ to $10^1$. Subsequently, with a fixed $\alpha$ of 1.0 or 0.1, we explore 50 sample sizes between 200 and 10,000. Both scenarios are run 100 times, calculating the mean squared error (MSE) between true ratios. For pseudo-training samples, we maintain a uniform class distribution and select batch sizes like 100, divisible by the number of classes. The choice of $\alpha$ for pseudo-test samples is informed by a normal distribution, defining the label distribution $p(\boldsymbol{y})$ and determining the number of samples per class after fixing $\alpha$. Experimental details and additional experiment are in Appendix H.

To evaluate relaxed label shift, we consider a distributional distance $\mathcal{D}$ and $\epsilon > 0$. Garg et al. (2023) define this as $\max_{\mathbf{y}} \mathcal{D}\left(p_{\mathrm{tr}}(\mathbf{x} \mid \mathbf{y}), p_{\mathrm{te}}(\mathbf{x} \mid \mathbf{y})\right) \leq \epsilon$. Instead of using temporally distinct datasets (Garg et al., 2023) like CIFAR-10 and CIFAR-10.1_v6 (Torralba et al., 2008; Recht et al., 2018), we employ a mild test data augmentation strategy to control the relaxation degree. To assess worst-case robustness in Eq. (3.5), we introduce variations in the pseudo-test sampling parameter $\alpha$ by using different means (-0.25, 0.0, 0.5) of the normal distribution for log_alpha selection.

A two-layer MLP is used on MNIST and ResNet-18 (He et al., 2016) on CIFAR-10 with post-hoc calibration same as MLLS. The kernels employed are linear for MNIST and RBF with a deviation of 0.25 for CIFAR-10.

Fig. 2 illustrates the comparable and superior performance of MLLS with convex optimization (MLLS_L1 and MLLS_L2 with convex solver) and the EM method (MLLS_EM), especially under severe shifts, relative to the gradient-based method (MLLS_CG with conjugate gradient descent). We later designate MLLS_EM and MLLS_L2 as baselines, and Fig. 3 shows our method consistently achieves lower MSE loss under various label shifts and sample sizes. Even under relaxed label shift conditions, our method outperforms the baselines, as shown in Fig. 4.

Lastly, ablation studies in Fig. 5 depicts the decline in performance with negative steering of the normal distribution of $\alpha$, indicating worsened scenarios and increased MSE.

**FL Settings.** We further apply VRLS in a FL context, addressing intra- and inter-client label shifts. Initially, experiments are conducted with 5 clients, utilizing predefined label distributions on Fashion MNIST (Xiao et al., 2017) and CIFAR-10, as illustrated in Table 7 and Table 8 (complete results in Appendix H). A sophisticated MLP with dropout serves as the predictor for Fashion MNIST. For the global training with IW-ERM, LeNet (LeCun et al., 1998) and ResNet-18 are adopted on Fashion MNIST and CIFAR-10 (Ramezani-Kebrya et al., 2023), respectively. We run experiments with three seeds, average the accuracy across clients, and compare IW-ERM with VRLS to FedAvg and FedBN baselines, as well as IW-ERM with the true ratios Upper Bound. All hyper-parameters are maintained consistent with those outlined in (McMahan et al., 2017; Li et al., 2021a; Ramezani-Kebrya et al., 2023).

Subsequently, we execute experiments involving 100 clients on CIFAR-10, selecting five clients randomly at each iteration to simulate a scenario closer to real-world FL. Here, IW-ERM with the true ratios does not act as the upper bound due to client sampling. The experiment is conducted once, and the average accuracy across clients is noted, with the label distribution detailed in Table 9.

Despite the reported slow convergence of FedBN in (Ramezani-Kebrya et al., 2023), for a fair comparison, we retain 15,000 and 10,000 iterations for both FedAvg and FedBN on Fashion MNIST and CIFAR-10, respectively, while limiting IW-ERM to 5,000 iterations with both true and our estimated ratios, owing to their rapid convergence.

Table 1 demonstrates that our IW-ERM surpasses the FedAvg and FedBN baselines by over 20% in average accuracy with only a third of the iterations on Fashion MNIST. Similarly, Table 2 reveals that our IW-ERM nearly reaches the upper bound on CIFAR-10, outperforming both baselines. The individual accuracy for each client is detailed in Table 5 and Table 6. In the 100-client scenario, our IW-ERM continues to exhibit superiority, requiring only half the iterations, as shown in Table 3. It is

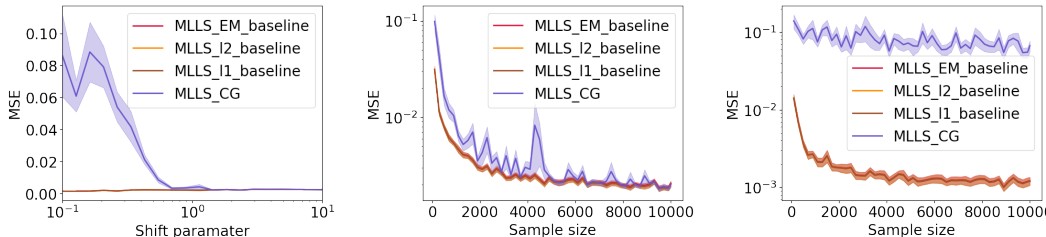

**Figure 2:** *MSE analysis on MNIST for MLLS baselines. **Left:** Performance evaluation across different alpha values, comparing the MLLS_EM, MLLS_L1, and MLLS_L2 methods with MLLS_CG with conjugate gradient descent. Both the EM and convex optimization methods demonstrate similar and superior performance compared to the gradient-based method, especially under severe label shift conditions. **Middle:** MSE analysis at an $\alpha$ of 1.0, highlighting the comparable performance of various methods, with the exception of MLLS_CG. **Right:** Analysis at $\alpha = 0.1$ illustrating that MLLS_CG is significantly inferior to both the EM and convex optimization methods, consistent with the left plot.*

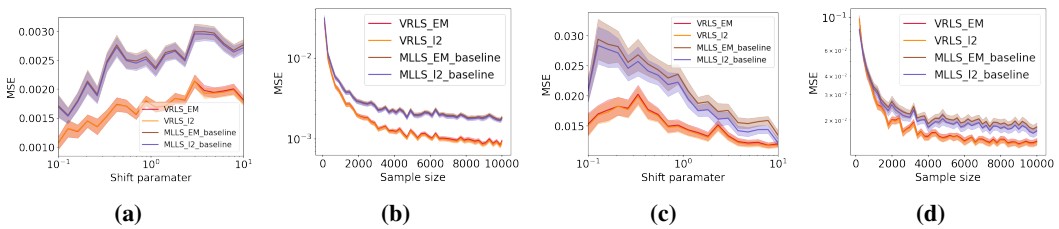

**Figure 3:** *MSE analysis on MNIST and CIFAR-10 for VRLS (ours) compared to baselines. Figs. 3a and 3c illustrate the consistent superiority of VRLS over MLLS baselines across varying alpha values on both datasets. Figs. 3b and 3d demonstrate VRLS performance improvement with different test sizes on both MNIST and CIFAR-10.*

noteworthy that the use of true ratios does not necessarily signify FTW-ERM, given the stochastic nature of client selection for training in each iteration.

**Table 1:** We employ LeNet on Fashion MNIST, addressing label shifts across 5 clients. For FedAvg, FedBN, FedProx and SCAFFOLD, we run 15,000 iterations, while running only 5,000 iterations for both Upper Bound (IW-ERM with true ratios) employing true ratios and our IW-ERM with VRLS. It is worth mentioning that, for training our predictor, we utilize a simple MLP with dropout and incorporate a linear kernel. Detailed results are documented in Table 5.

| Method | Avg.accuracy |
|---|---|
| **Our IW-ERM** | **$0.7520 \pm 0.0209$** |
| FedAvg | $0.5472 \pm 0.0297$ |
| FedBN | $0.5359 \pm 0.0306$ |
| FedProx | $0.5606 \pm 0.0070$ |
| SCAFFOLD | $0.5774 \pm 0.0036$ |
| Upper Bound | $0.8273 \pm 0.0041$ |

# 6 CONCLUSIONS AND FUTURE WORK

We have addressed intra- and inter-client label shifts as a major challenge in FL by developing an efficient and privacy-preserving IW-ERM. Our density ratio estimation incorporates regularization di-

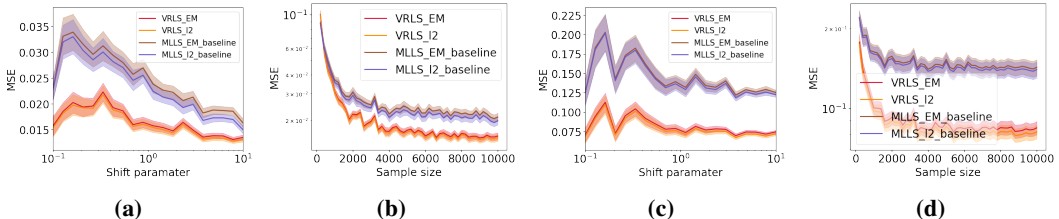

(a)      (b)      (c)      (d)

**Figure 4:** *Extended MSE analysis on CIFAR-10 illustrating the impact of simulated **relaxed label shift** conditions effected through slight data augmentations applied to the test dataset. Figs. 4a and 4b test data augmentation with a 30% chance of occurrence: Gaussian blur (kernel size: 3; σ: 0.1–0.5) and brightness adjustment (factor: ±0.1), instilling a degree of real-world variability. Figs. 4c and 4d showcase a 50% probability of similar augmentation but with attenuated parameters for blur (σ: 0.1–0.7) and brightness (factor: ±0.2), exploring the impacts of more subtle adjustments.*

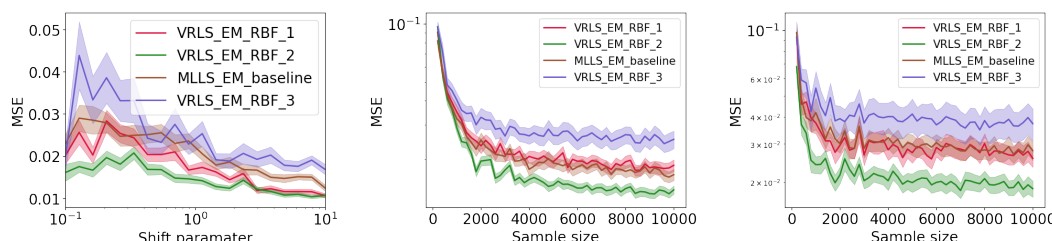

**Figure 5:** *Ablation studies by varying α's and test sizes on CIFAR-10 for VRLS (ours). **Left:** The analysis conducted under three scenarios differentiated by the means of the Dirichlet parameter log_alphas sampled with the same RBF configuration: RBF_1 (mean=0.0, std=0.25), RBF_2 (mean=-0.25, std=0.25), and RBF_3 (mean=0.5, std=0.25), suggesting a negative mean direction enhances performance. **Center:** We assess performance across varying test sizes at $\alpha = 1.0$, using three different log_alpha setups. The results indicate a performance improvement trajectory from RBF_3 to RBF_1, culminating in the highest performance at RBF_2. **Right:** Analysis conducted at $\alpha = 0.1$, illustrating the same trend observed at $\alpha = 1.0$.*

**Table 2:** We deploy ResNet-18 on CIFAR-10 to handle label shift across 5 clients. The predictor is also a ResNet-18, equipped with an RBF kernel, maintaining consistency with the single-client scenario. To ensure a fair comparison, we conduct 5,000 iterations for IW-ERM with VRLS and true ratios, while running 10,000 iterations for both FedAvg and FedBN. Detailed results are documented in Table 6.

| CIFAR-10 | Our IW-ERM | FedAvg | FedBN | Upper Bound |
|---|---|---|---|---|
| **Avg. accuracy** | **0.5640 ± 0.0241** | 0.4515 ± 0.0148 | 0.4263 ± 0.0975 | 0.5790 ± 0.0103 |

**Table 3:** We present the average client accuracies from the CIFAR-10 target shift experiment conducted across 100 and 200 clients, with 5 clients randomly sampled to participate in each training round. Our IW-ERM with VRLS runs for 5,000 and 10,000 iterations individually, while FedAvg and FedBN each runs 10,000 iterations.

| CIFAR-10 | Our IW-ERM | FedAvg | FedBN |
|---|---|---|---|
| **Avg. accuracy (100 clients)** | **0.5354** | 0.3915 | 0.1537 |
| **Avg. accuracy (200 clients)** | **0.6216** | 0.5942 | 0.1753 |

rectly into the predictor training process along with a similarity comparison task designed specifically for label shift assumptions. Our empirical results highlight significant performance improvements of our VRLS compared to the MLLS family baselines across several datasets in a single-client scenario as well as the superiority over all FL baselines. Overall, our VRLS with IW-ERM presents a significant advancement in addressing label shifts challenges in FL.

## 7 ETHICS STATEMENT

Developing privacy-preserving learning algorithms forms a cornerstone of responsible and ethical AI practices. Our work addresses FL by developing innovative algorithms designed to potentially eliminate the risk of data leakage with importance weighting. However, the long-term implications of our schemes remain contingent upon the manner in which machine learning is implemented and utilized within society.

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

The supplementary part is organized as follows:

- All related work are provided in Appendix A.
- Additional details of prior work of BBSE and MLLS are in Appendix B.
- Mathematical proof for label shifts with multiple clients and IW-ERM is given in Appendix C.
- General algorithmic description is in Appendix D.
- Proof of Theorem 1 is in Appendix E.
- Proof of Theorem 2 and Convergence-Communication-Privacy guarantees for Eq. (IW-ERM) are provided in Appendix F.
- We provide the comparison of Latent Distribution in Appendix G.
- Additional experiments and experimental details are provided in Appendix H.
- Complexity analysis is in Appendix I.
- Mathematical notations are summarized in Appendix J.
- Limitations are discussed in Appendix K.

## A  RELATED WORK

In the context of FL with label shifts, importance ratio estimation is tackled either by solving a linear system as in (Lipton et al., 2018; Azizzadenesheli et al., 2019) or by minimizing distribution divergence as in (Garg et al., 2020). In this section, we overview complete related work.

**Federated learning.**  Much of the current research in FL predominantly centers around the minimization of empirical risk, operating under the assumption that each client maintains the same training/test data distribution (Li et al., 2020; Kairouz et al., 2021; Wang et al., 2021). Prominent methods in FL (Kairouz et al., 2021; Li et al., 2020; Wang et al., 2021) include FedAvg (McMahan et al., 2017) and FedBN (Li et al., 2021a). FedAvg and its variants such as (Huang et al., 2021; Karimireddy et al., 2020) have been the subject of thorough investigation in optimization literature, exploring facets such as communication efficiency, client participation, and privacy assurance (Ramezani-Kebrya et al., 2023).Subsequent work, such as the study by de Luca et al. (2022), explores Federated Domain Generalization and introduces data augmentation to the training. This model aims to generalize to both in-domain datasets from participating clients and an out-of-domain dataset from a non-participating client. Additionally, Gupta et al. (2022) introduces FL Games, a game-theoretic framework designed to learn causal features that remain invariant across clients. This is achieved by employing ensembles over clients' historical actions and enhancing local computation, under the assumption of consistent training/test data distribution across clients. The existing strategies to address statistical heterogeneity across clients during training primarily rely on heuristic-based personalization methods, which currently lack theoretical backing in statistical learning (Smith et al., 2017; Khodak et al., 2019; Li et al., 2021b). In contrast, we aim to minimize overall test error amid both intra-client and inter-client distribution shifts, a situation frequently observed in real-world scenarios. Techniques ensuring communication efficiency, robustness, and secure aggregations serve as complementary.

**Importance ratio estimation**  Classical Empirical Risk Minimization (ERM) seeks to minimize the expected loss over the training distribution using finite samples. When faced with distribution shifts, the goal shifts to minimizing the expected loss over the target distribution, leading to the development of Importance-Weighted Empirical Risk Minimization (IW-ERM)(Shimodaira, 2000; Sugiyama et al., 2006; Byrd and C. Lipton, 2019; Fang et al., 2020). Shimodaira (2000) established that the IW-ERM estimator is asymptotically unbiased. Moreover, Ramezani-Kebrya et al. (2023) introduced FTW-ERM, which integrates density ratio estimation.

**Label shift and MLLS family**  For theoretical analysis, the conditional distribution $p(\boldsymbol{x}|\boldsymbol{y})$ is held strictly constant across all distributions (Lipton et al., 2018; Garg et al., 2020; Saerens et al., 2002). Both BBSE (Lipton et al., 2018) and RLLS (Azizzadenesheli et al., 2019) designate a discrete latent

space $z$ and introduce a confusion matrix-based estimation method to compute the ratio $w$ by solving a linear system (Saerens et al., 2002; Lipton et al., 2018). This approach is straightforward and has been proven consistent, even when the predictor is not calibrated. However, its subpar performance is attributed to the information loss inherent in the confusion matrix (Garg et al., 2020).

Consequently, MLLS (Garg et al., 2020) introduces a continuous latent space, resulting in a significant enhancement in estimation performance, especially when combined with a post-hoc calibration method (Shrikumar et al., 2019). It also provides a consistency guarantee with a canonically calibrated predictor. This EM-based MLLS method is both concave and can be solved efficiently.

**Discrepancy Measure**    In information theory and statistics, discrepancy measures play a critical role in quantifying the differences between probability distributions. One such measure is the Bregman Divergence (Banerjee et al., 2005), defined as

$$D_\phi(\boldsymbol{x}\|\boldsymbol{y}) = \phi(\boldsymbol{x}) - \phi(\boldsymbol{y}) - \langle \nabla\phi(\boldsymbol{y}), \boldsymbol{x} - \boldsymbol{y}\rangle,$$

which encapsulates the difference between the value of a convex function $\phi$ at two points and the value of the linear approximation of $\phi$ at one point, leveraging the gradient at another point.

Discrepancy measures are generally categorized into two main families: Integral Probability Metrics (IPMs) and $f$-divergences. IPMs, including Maximum Mean Discrepancy (Gretton et al., 2012) and Wasserstein distance (Villani, 2009), focus on distribution differences $P - Q$. In contrast, $f$-divergences, such as KL-divergence (Kullback and Leibler, 1951) and Total Variation distance, operate on ratios $P/Q$ and do not satisfy the triangular inequality. Interconnections and variations between these families are explored in studies like $(f, \Gamma)$-Divergences (Birrell et al., 2022), which interpolate between $f$-divergences and IPMs, and research outlining optimal bounds between them (Agrawal and Horel, 2020).

MLLS (Garg et al., 2020) employs $f$-divergence, notably the KL divergence, which is not a metric as it doesn't satisfy the triangular inequality, and requires distribution $P$ to be absolutely continuous with respect to $Q$. Concerning IPMs, while MMD is reliant on a kernel function, it can suffer from the curse of dimensionality when faced with high-dimensional data. On the other hand, the Wasserstein distance can be reformulated using Kantorovich-Rubinstein duality (Dedecker et al., 2006; Arjovsky et al., 2017) as a maximization problem subject to a Lipschitz constrained function $f : \mathbb{R}^d \to \mathbb{R}$.

# B    BBSE AND MLLS FAMILY

In this part, we conclude the contributions from (Lipton et al., 2018; Garg et al., 2020). Our goal is to estimate the ratio $p^{\text{te}}(\mathbf{y})/p^{\text{tr}}(\mathbf{y})$. In our setup, we set $|\mathcal{Y}| = m$ since we have $m$ possible label classes, denoted as $y = i$, for $i \in [m]$ and $[m] = \{1, 2, \ldots, m\}$. Let $\mathbf{r}^\star = [r_1^\star, \ldots, r_m^\star]^\top$ denote the true ratios. Each $r_i^\star$ is defined as $r_i^\star = p^{\text{te}}(y = i)/p^{\text{tr}}(y = i)$ (Garg et al., 2020). Then we first define a a family of distributions over $\mathcal{Z}$ parameterized by $\mathbf{r} = [r_1, \ldots, r_m]^\top \in \mathbb{R}^m$, with $r_i$ being the $i$-th element:

$$p_{\mathbf{r}}(\mathbf{z}) := \sum_{i=1}^{m} p^{\text{te}}(\mathbf{z}|y = i) \cdot p^{\text{tr}}(y = i) \cdot r_i \tag{B.1}$$

where $r_i \geq 0$ for $i \in [m]$ and $\sum_{i=1}^{m} r_i \cdot p^{\text{tr}}(y = i) = \sum_{i=1}^{m} p^{\text{te}}(y = i) = 1$ as constraints. When $\mathbf{r} = \mathbf{r}^\star$ ($r_i = r_i^\star$ for $i \in [m]$), we have $p_{\mathbf{r}}(\mathbf{z}) = p_{\mathbf{r}^\star}(\mathbf{z}) = p^{\text{te}}(\mathbf{z})$ Garg et al. (2020). So our task is to find $\mathbf{r}$ such that

$$\sum_{i=1}^{m} p^{\text{te}}(\mathbf{z}|y = i) \cdot p^{\text{tr}}(y = i) \cdot r_i = \sum_{i=1}^{m} p^{\text{tr}}(\mathbf{z}, y = i) \cdot r_i = p^{\text{te}}(\mathbf{z}) \tag{B.2}$$

Lipton et al. (2018); Garg et al. (2020) proposed Black Box Shift Estimation (BBSE) to solve this problem. With a pre-defined predictor $f$ trained on classification task, this approach assumes $\mathcal{Z}$ to be discrete latent space and set $p(\mathbf{z}|\mathbf{x}) = \delta_{\arg\max f(\mathbf{x})}$ where the output of $f(\mathbf{x})$ is $m$-digit simplex. BBSE estimates the $p^{\text{te}}(\mathbf{z}|\mathbf{y})$ as a confusion matrix using training and validation data, $p^{\text{tr}}(y = i)$ with training set and $p^{\text{te}}(\mathbf{z})$ with test data. Then we just need to solve the equation:

$$\mathbf{A}\mathbf{w} = \mathbf{B} \tag{B.3}$$

where $|\mathcal{Z}| = m$, $\mathbf{A} \in \mathbb{R}^{m \times m}$ with $\mathbf{A}_{ji} = p^{\text{te}}(z = j|y = i) \cdot p^{\text{tr}}(y = i)$, and $\mathbf{B} \in \mathbb{R}^m$ with $\mathbf{B}_j = p^{\text{te}}(z = j)$ for $i \in [m]$ and $j \in [m]$.

The estimation of the confusion matrix in terms of $p^{\text{te}}(\mathbf{z}|\mathbf{y})$ leads to the loss of calibration information (Garg et al., 2020). Furthermore, when defining $\mathcal{Z}$ as a continuous latent space, the confusion matrix becomes intractable since $\mathbf{z}$ has infinitely many values. Therefore, MLLS directly minimizes the divergence between $p^{\text{te}}(\mathbf{z})$ and $p_{\mathbf{r}}(\mathbf{z})$, instead of solving the linear system Eq. (B.3).

Within the $f$-divergence family, MLLS seeks to find a weight vector $\mathbf{r}$ by minimizing the KL-divergence $\text{KL}\left(p^{\text{te}}(\mathbf{z}), p_{\mathbf{r}}(\mathbf{z})\right) = \mathbb{E}_{\text{te}}\left[\log p^{\text{te}}(\mathbf{z})/p_{\mathbf{r}}(\mathbf{z})\right]$, for $p_{\mathbf{r}}(\mathbf{z})$ defined in Eq. (B.1). Leveraging on the properties of the logarithm, this is equivalent to maximizing the log-likelihood: $\mathbf{r} := \arg\max_{\mathbf{r} \in \mathbb{R}} \mathbb{E}_{\text{te}}\left[\log p_{\mathbf{r}}(\mathbf{z})\right]$. Expanding $p_{\mathbf{r}}(\mathbf{z})$, we have

$$
\begin{aligned}
\mathbb{E}_{\text{te}}\left[\log p_{\mathbf{r}}(\mathbf{z})\right] &= \mathbb{E}_{\text{te}}\left[\log\left(\sum_{i=1}^{m} p^{\text{tr}}(\mathbf{z}, y = i) r_i\right)\right] \\
&= \mathbb{E}_{\text{te}}\left[\log\left(\sum_{i=1}^{m} p^{\text{tr}}(y = i \mid \mathbf{z}) r_i\right) + \log p^{\text{tr}}(\mathbf{z})\right].
\end{aligned}
\tag{B.4}
$$

Therefore the unified form of MLLS can be formulated as:

$$
\mathbf{r} := \arg\max_{\mathbf{r} \in \mathbb{R}} \mathbb{E}_{\text{te}}\left[\log\left(\sum_{i=1}^{m} p^{\text{tr}}(y = i \mid \mathbf{z}) r_i\right)\right].
\tag{B.5}
$$

This is a convex optimization problem and can be solved efficiently using methods such as EM, an analytic approach, and also iterative optimization methods like gradient descent with labeled training data and unlabeled test data. MLLS defines the $p(\mathbf{z}|\mathbf{x})$ as $\delta_{\mathbf{x}}$, plugs in the pre-defined $f$ to approximate $p^{\text{tr}}(\mathbf{y}|\mathbf{x})$ and optimizes the following objective:

$$
\mathbf{r}_f := \arg\max_{\mathbf{r} \in \mathbb{R}} \ell(\mathbf{r}, f) := \arg\max_{\mathbf{r} \in \mathbb{R}} \mathbb{E}_{\text{te}}\left[\log(f(\mathbf{x})^T \mathbf{r})\right].
\tag{B.6}
$$

With the Bias-Corrected Calibration (BCT) (Shrikumar et al., 2019) strategy, they adjust the logits $\hat{f}(\mathbf{x})$ of $f(\mathbf{x})$ element-wise for each class, and the objective becomes:

$$
\mathbf{r}_f := \arg\max_{\mathbf{r} \in \mathbb{R}} \ell(\mathbf{r}, f) := \arg\max_{\mathbf{r} \in \mathbb{R}} \mathbb{E}_{\text{te}}\left[\log(g \circ \hat{f}(\mathbf{x}))^T \mathbf{r})\right],
\tag{B.7}
$$

where $g$ is a calibration function.

By defining $p(\mathbf{z}|\mathbf{x})$ as $\delta_{\arg\max f(\mathbf{x})}$ and plugging in the predictor $f$ as well as the post-hoc calibration term, we derive the objective of MLLS-CM:

$$
\mathbf{r}_f := \arg\max_{\mathbf{r} \in \mathbb{R}} \ell(\mathbf{r}, f) := \arg\max_{\mathbf{r} \in \mathbb{R}} \mathbb{E}_{\text{te}}\left[\log((g \circ \hat{f}(x))C)^T \mathbf{r})\right].
\tag{B.8}
$$

where $C \in R^{m \times m}$ is the confusion matrix approximator of $p^{\text{tr}}(\hat{f}(\mathbf{x})|\mathbf{y})$. MLLS-CM can be implemented using EM with a transformed predictor $(g \circ \hat{f}(\mathbf{x}))C$ instead of $g \circ \hat{f}(\mathbf{x})$ in MLLS.

## C  PROOF OF PROPOSITION 1

In the following, we consider four typical scenarios under various distributions shifts and formulate their IW-ERM with a focus on minmizing $R_1$.

**Table 4:** Details of scenarios described in Section 2

| Scenario | #Clients | Assumptions on Distributions | Ratio Client i Needs |
|---|---|---|---|
| `No-LS` in (C.1) | 2 | $p_1^{\text{tr}}(\mathbf{y}) = p_1^{\text{te}}(\mathbf{y})$ and $p_1^{\text{tr}}(\mathbf{y}) \neq p_2^{\text{tr}}(\mathbf{y})$ | $p_1^{\text{tr}}(\mathbf{y})/p_2^{\text{tr}}(\mathbf{y})$ |
| `LS on single` in (C.2) | 2 | $p_1^{\text{tr}}(\mathbf{y}) \neq p_1^{\text{te}}(\mathbf{y})$ and $p_2^{\text{tr}}(\mathbf{y}) = p_2^{\text{te}}(\mathbf{y})$ | $p_1^{\text{te}}(\mathbf{y})/p_1^{\text{tr}}(\mathbf{y})$ and $p_1^{\text{te}}(\mathbf{y})/p_2^{\text{tr}}(\mathbf{y})$ |
| `LS on both` in (C.2) | 2 | $p_1^{\text{tr}}(\mathbf{y}) \neq p_1^{\text{te}}(\mathbf{y})$ and $p_2^{\text{tr}}(\mathbf{y}) \neq p_2^{\text{te}}(\mathbf{y})$ | $p_1^{\text{te}}(\mathbf{y})/p_1^{\text{tr}}(\mathbf{y})$ and $p_1^{\text{te}}(\mathbf{y})/p_2^{\text{tr}}(\mathbf{y})$ |
| `LS on multi` in (C.3) | $K$ | $p_k^{\text{tr}}(\mathbf{y}) \neq p_1^{\text{te}}(\mathbf{y})$ for all $k$ | $p_1^{\text{te}}(\mathbf{y})/p_k^{\text{tr}}(\mathbf{y})$ for all $k$ |

## C.1 NO INTRA-CLIENT LABEL SHIFT

For description simplicity, we assume that there are only 2 clients but our results can be directly extended to multiple clients. This scenario assumes $p_k^{\text{tr}}(\mathbf{y}) = p_k^{\text{te}}(\mathbf{y})$ for $k = 1, 2$, but $p_1^{\text{tr}}(\mathbf{y}) \neq p_2^{\text{tr}}(\mathbf{y})$. Client 1 aims to learn $h_{\mathbf{w}}$ assuming $\frac{p_1^{\text{tr}}(\mathbf{y})}{p_2^{\text{tr}}(\mathbf{y})}$ is given. We consider the following IW-ERM that is proved to be consistent in terms of minimizing $R_1$:

$$\min_{h_{\mathbf{w}} \in \mathcal{H}} \frac{1}{n_1^{\text{tr}}} \sum_{i=1}^{n_1^{\text{tr}}} \ell(h_{\mathbf{w}}(\mathbf{x}_{1,i}^{\text{tr}}), \mathbf{y}_{1,i}^{\text{tr}}) + \frac{1}{n_2^{\text{tr}}} \sum_{i=1}^{n_2^{\text{tr}}} \frac{p_1^{\text{tr}}(\mathbf{y}_{2,i}^{\text{tr}})}{p_2^{\text{tr}}(\mathbf{y}_{2,i}^{\text{tr}})} \ell(h_{\mathbf{w}}(\mathbf{x}_{2,i}^{\text{tr}}), \mathbf{y}_{2,i}^{\text{tr}}). \tag{C.1}$$

Here $\mathcal{H}$ is the hypothesis class of $h_{\mathbf{w}}$. The scenario is short for `No-LS`.

## C.2 LABEL SHIFT ONLY FOR CLIENT 1

(short for `LS on single`) Here we consider label shift only for client 1, i.e., $p_1^{\text{tr}}(\mathbf{y}) \neq p_1^{\text{te}}(\mathbf{y})$ and $p_2^{\text{tr}}(\mathbf{y}) = p_2^{\text{te}}(\mathbf{y})$. We consider the following IW-ERM:

$$\min_{h_{\mathbf{w}} \in \mathcal{H}} \frac{1}{n_1^{\text{tr}}} \sum_{i=1}^{n_1^{\text{tr}}} \frac{p_1^{\text{te}}(\mathbf{y}_{1,i}^{\text{tr}})}{p_1^{\text{tr}}(\mathbf{y}_{1,i}^{\text{tr}})} \ell(h_{\mathbf{w}}(\mathbf{x}_{1,i}^{\text{tr}}), \mathbf{y}_{1,i}^{\text{tr}}) + \frac{1}{n_2^{\text{tr}}} \sum_{i=1}^{n_2^{\text{tr}}} \frac{p_1^{\text{te}}(\mathbf{y}_{2,i}^{\text{tr}})}{p_2^{\text{tr}}(\mathbf{y}_{2,i}^{\text{tr}})} \ell(h_{\mathbf{w}}(\mathbf{x}_{2,i}^{\text{tr}}), \mathbf{y}_{2,i}^{\text{tr}}). \tag{C.2}$$

This scenario is short for `LS on single`.

## C.3 LABEL SHIFT FOR BOTH CLIENTS

Here we assume $p_1^{\text{tr}}(\mathbf{y}) \neq p_1^{\text{te}}(\mathbf{y})$ and $p_2^{\text{tr}}(\mathbf{y}) \neq p_2^{\text{te}}(\mathbf{y})$, *i.e.,* label shift for both clients. The corresponding IW-ERM is the same as Eq. (C.2). This scenario is short for `LS on both`.

Without loss of generality and for simplicity of notation, in this section, we set $l = 1$. We consider four typical scenarios under various distribution shifts and formulate their IW-ERM with a focus on minmizing $R_1$. The details of these scenarios are summarized in Table 4.

## C.4 MULTIPLE CLIENTS

Here we consider a general scenario with $K$ clients. We assume both intra-client and inter-client label shifts by the following IW-ERM:

$$\min_{h_{\mathbf{w}} \in \mathcal{H}} \sum_{k=1}^{K} \frac{\lambda_k}{n_k^{\text{tr}}} \sum_{i=1}^{n_k^{\text{tr}}} \frac{p_1^{\text{te}}(\mathbf{y}_{k,i}^{\text{tr}})}{p_k^{\text{tr}}(\mathbf{y}_{k,i}^{\text{tr}})} \ell(h_{\mathbf{w}}(\mathbf{x}_{k,i}^{\text{tr}}), \mathbf{y}_{k,i}^{\text{tr}}), \tag{C.3}$$

where $\sum_{k=1}^{K} \lambda_k = 1$ and $\lambda_k \geq 0$. This scenario is short for `LS on multi`.

For the scenario without intra-client label shift, the IW-ERM in Eq. (C.1) can be expressed as

$$
\begin{aligned}
\frac{1}{n_2^{\mathrm{tr}}} \sum_{i=1}^{n_2^{\mathrm{tr}}} \frac{p_1^{\mathrm{tr}}(\mathbf{y}_{2,i}^{\mathrm{tr}})}{p_2^{\mathrm{tr}}(\mathbf{y}_{2,i}^{\mathrm{tr}})} \ell(h_{\mathbf{w}}(\mathbf{x}_{2,i}^{\mathrm{tr}}), \mathbf{y}_{2,i}^{\mathrm{tr}}) & \xrightarrow{n_2^{\mathrm{tr}} \to \infty} \mathbb{E}_{p_2^{\mathrm{tr}}(\mathbf{x},\mathbf{y})} \left[ \frac{p_1^{\mathrm{tr}}(\mathbf{y})}{p_2^{\mathrm{tr}}(\mathbf{y})} \ell(h_{\mathbf{w}}(\mathbf{x}), \mathbf{y}) \right] \\
& = \int_{\mathcal{Y}} \frac{p_1^{\mathrm{tr}}(\mathbf{y})}{p_2^{\mathrm{tr}}(\mathbf{y})} \mathbb{E}_{p(\mathbf{x}|\mathbf{y})}[\ell(h_{\mathbf{w}}(\mathbf{x}), \mathbf{y})] p_2^{\mathrm{tr}}(\mathbf{y}) \, \mathrm{d}\mathbf{y} \\
& = \int_{\mathcal{Y}} p_1^{\mathrm{tr}}(\mathbf{y}) \mathbb{E}_{p(\mathbf{x}|\mathbf{y})}[\ell(h_{\mathbf{w}}(\mathbf{x}), \mathbf{y})] \, \mathrm{d}\mathbf{y} \\
& = \int_{\mathcal{Y}} p_1^{\mathrm{te}}(\mathbf{y}) \mathbb{E}_{p(\mathbf{x}|\mathbf{y})}[\ell(h_{\mathbf{w}}(\mathbf{x}), \mathbf{y})] \, \mathrm{d}\mathbf{y} \\
& = \mathbb{E}_{p_1^{\mathrm{te}}(\mathbf{x},\mathbf{y})} [\ell(h_{\mathbf{w}}(\mathbf{x}), \mathbf{y})] \\
& = R_1(h_{\mathbf{w}}).
\end{aligned}
\tag{C.4}
$$

where the second equality holds due the assumption of the label shift setting and Bayes' theorem: $p(\mathbf{x}, \mathbf{y}) = p(\mathbf{x}|\mathbf{y}) \cdot p(\mathbf{y})$, the fourth equality holds by the assumption that $p_1^{\mathrm{tr}}(\mathbf{y}) = p_1^{\mathrm{te}}(\mathbf{y})$ in No-LS setting.

For the scenario with label shift only for Client 1 or for both clients, the IW-ERM in Eq. (C.2) admits

$$
\begin{aligned}
\frac{1}{n_2^{\mathrm{tr}}} \sum_{i=1}^{n_2^{\mathrm{tr}}} \frac{p_1^{\mathrm{te}}(\mathbf{y}_{2,i}^{\mathrm{tr}})}{p_2^{\mathrm{tr}}(\mathbf{y}_{2,i}^{\mathrm{tr}})} \ell(h_{\mathbf{w}}(\mathbf{x}_{2,i}^{\mathrm{tr}}), \mathbf{y}_{2,i}^{\mathrm{tr}}) & \xrightarrow{n_2^{\mathrm{tr}} \to \infty} \mathbb{E}_{p_2^{\mathrm{tr}}(\mathbf{x},\mathbf{y})} \left[ \frac{p_1^{\mathrm{te}}(\mathbf{y})}{p_2^{\mathrm{tr}}(\mathbf{y})} \ell(h_{\mathbf{w}}(\mathbf{x}), \mathbf{y}) \right] \\
& = \int_{\mathcal{Y}} \frac{p_1^{\mathrm{te}}(\mathbf{y})}{p_2^{\mathrm{tr}}(\mathbf{y})} \mathbb{E}_{p(\mathbf{x}|\mathbf{y})}[\ell(h_{\mathbf{w}}(\mathbf{x}), \mathbf{y})] p_2^{\mathrm{tr}}(\mathbf{y}) \, \mathrm{d}\mathbf{y} \\
& = \int_{\mathcal{Y}} p_1^{\mathrm{te}}(\mathbf{y}) \mathbb{E}_{p(\mathbf{x}|\mathbf{y})}[\ell(h_{\mathbf{w}}(\mathbf{x}), \mathbf{y})] \, \mathrm{d}\mathbf{y} \\
& = \mathbb{E}_{p_1^{\mathrm{te}}(\mathbf{x},\mathbf{y})} [\ell(h_{\mathbf{w}}(\mathbf{x}), \mathbf{y})] \\
& = R_1(h_{\mathbf{w}}).
\end{aligned}
$$

For multiple clients, let $k \in [K]$. Similarly, we have

$$
\frac{1}{n_k^{\mathrm{tr}}} \sum_{i=1}^{n_k^{\mathrm{tr}}} \frac{p_1^{\mathrm{te}}(\mathbf{y}_{k,i}^{\mathrm{tr}})}{p_k^{\mathrm{tr}}(\mathbf{y}_{k,i}^{\mathrm{tr}})} \ell(h_{\mathbf{w}}(\mathbf{x}_{k,i}^{\mathrm{tr}}), \mathbf{y}_{k,i}^{\mathrm{tr}}) \xrightarrow{n_k^{\mathrm{tr}} \to \infty} R_1(h_{\mathbf{w}}).
$$

Then we have

$$
\sum_{k=1}^{K} \frac{\lambda_k}{n_k^{\mathrm{tr}}} \sum_{i=1}^{n_k^{\mathrm{tr}}} \frac{p_1^{\mathrm{te}}(\mathbf{y}_{k,i}^{\mathrm{tr}})}{p_k^{\mathrm{tr}}(\mathbf{y}_{k,i}^{\mathrm{tr}})} \ell(h_{\mathbf{w}}(\mathbf{x}_{k,i}^{\mathrm{tr}}), \mathbf{y}_{k,i}^{\mathrm{tr}}) \xrightarrow{n_1^{\mathrm{tr}}, \dots, n_K^{\mathrm{tr}} \to \infty} R_1(h_{\mathbf{w}}).
$$

Note that to solve Eq. (C.3), client 1 needs to estimate $\frac{p_1^{\mathrm{te}}(\mathbf{y})}{p_k^{\mathrm{tr}}(\mathbf{y})}$ for all clients $k$ with $\lambda_k > 0$ in (C.3).

The consistency of Eq. (IW-ERM), i.e., convergence in probability, is followed the standard arguments in e.g., (Shimodaira, 2000)[Section 3] and (Sugiyama et al., 2007)[Section 2.2] using the law of large numbers.

## D   ALGORITHMIC DESCRIPTION

```
1  # Split the training dataset on each client
2  trainsets = target_shift.split_dataset(trainset.data, trainset.targets,
       client_label_dist_train, transform=transform_train)
3
4  # Split the test dataset on each client
5  testsets = target_shift.split_dataset(testset.data, testset.targets,
       client_label_dist_test, transform=transform_test)
6
```

---

**Algorithm 1** Variational Regularized Label Shift (VRLS) Density Estimation

---

1: **Initialization:**
   - Initialize the predictor $f$.
   - Select an appropriate kernel.
2: **Data Preparation:**
   - Arrange the training data and establish the training loader.
   - **Pseudo-training Data:**
     - Sample uniformly across classes from training data.
     - Set up the pseudo-training mini-batch.
   - **Pseudo-test Data:**
     - Sample a Dirichlet distribution parameter $\alpha$.
     - Sample marginal label distributions.
     - Sample from training data and set up the pseudo-test mini-batch.
3: **Training Loop:**
   - Compute the cross-entropy loss.
   - Compute the MMD loss between the predictions for pseudo-training and pseudo-test data.
   - Optimize $f$ using both losses until the cross-entropy loss falls beneath a threshold or the maximum number of epochs is reached.
4: **Output:** The predictor $f$.
5: **Label Shift Estimation:**
   - Utilize $f$ and the unlabeled test data to address the Eq. (3.2) objective via methods such as the EM algorithm or convex optimization.
6: **Final Output:** Estimated label shift ratio.

---

---

**Algorithm 2** IW-ERM with VRLS for FL

---

1: **Initialization:** Initialize the global model.
2: **Data Preparation:** Distribute labeled training and unlabeled test data among local clients.
3: **Label Shift Ratio Estimation:**
   - **Predictor Training: Method A or B**
     - **Method A: Distributed Training (preferred)**
       * Train a global predictor in a distributed manner.
       * Broadcast the global model parameters to each client.
     - **Method B: Centralized Training (default)**
       * Each client uploads its training data to the server.
       * Train the predictor on the server.
       * Broadcast the predictor parameters to clients.
   - **Inference:**
     - Perform inference in each client to estimate the test label marginal distributions and send them to the server.
     - Calculate the training label marginal distribution locally in each client.
   - The server broadcasts the received distributions to all clients.
   - Each client calculates the sum of all test label distributions.
4: **Computing Ratios:** Compute ratios $\mathbf{r}_k$ on each client.
5: **IW-ERM:** Each client applies its own ratio $\mathbf{r}_k$ for IW-ERM to adjust its local model.
6: **Output:** Optimized global model.

---

```
7  # Training of the predictor, output net and calibration
8  net, calibration = initialize_fmnist()
9
10 # Initilize the estimator
11 estimator = LS_RatioModel(net, calibration)
12
13 # Initialize a tensor to store the estimated values for each client.
14 estimated_values = torch.zeros(client_num, nclass)
15
16 # Iterate through each client's testset to calculate the estimated values
        locally.
17
18 for i, testset in enumerate(testsets):
19     estimated_values[i] = estimator(testset.data.cpu().numpy()/255.0)
20
21 # Broadcast and sum of all test label marginal distributions on each
        client
22 estimated_values = torch.sum(estimated_values, dim=0, keepdim=True)
23
24 # Initialize a tensor to store the marginal value for each client
25 marginal_values = torch.zeros(client_num, nclass)
26
27 # Iterate through each client's trainset to calculate the marginal value
28 for i, trainset in enumerate(trainsets):
29     marginal_values[i] = marginal(trainset.targets)
30
31 # Obtain the ratios
32 ratios = (estimated_values / marginal_values).to(args.device)
```

**Listing 1:** Our VRLS in FL. Here, client_label_dist_train and client_label_dist_test are predefined. The estimated_values are estimated locally and the dimensionality is only clients_num*nclass.

## E  PROOF OF THEOREM 1

*Proof.* Let $H(\mathbf{r}, \boldsymbol{\theta}, \mathbf{x}) = -\log(f(\mathbf{x}, \boldsymbol{\theta})^\top \mathbf{r})$. From the strong convexity Lemma 3, we have that

$$\|\hat{\mathbf{r}}_n - \mathbf{r}_{f^\star}\|_2^2 \leq \frac{2}{\mu p_{\min}} \left( \mathcal{L}_{\boldsymbol{\theta}^\star}(\hat{\mathbf{r}}_n) - \mathcal{L}_{\boldsymbol{\theta}^\star}(\mathbf{r}_{f^\star}) \right) \tag{E.1}$$

Now focusing on the term on the right hand side, we find by invoking Lemma 4 that

$$\mathcal{L}_{\boldsymbol{\theta}^\star}(\hat{\mathbf{r}}_n) - \mathcal{L}_{\boldsymbol{\theta}^\star}(\mathbf{r}_{f^\star}) \leq \mathbb{E}\left[ H(\hat{\mathbf{r}}_n, \hat{\boldsymbol{\theta}}_n, \mathbf{x}) \right] - \mathbb{E}\left[ H(\mathbf{r}_{f^\star}, \hat{\boldsymbol{\theta}}_n, \mathbf{x}) \right] + 2L\mathbb{E}\left[ \|\hat{\boldsymbol{\theta}}_n - \boldsymbol{\theta}^\star\|_2 \right]$$

$$= \mathbb{E}\left[ H(\hat{\mathbf{r}}_n, \hat{\boldsymbol{\theta}}_n, x) \right] - \frac{1}{n}\sum_{i=1}^{n} H(\hat{\mathbf{r}}_n, \hat{\boldsymbol{\theta}}_n, \mathbf{x}_i) + \frac{1}{n}\sum_{i=1}^{n} H(\hat{\mathbf{r}}_n, \hat{\boldsymbol{\theta}}_n, \mathbf{x}_i)$$

$$- \mathbb{E}\left[ H(\mathbf{r}_{f^\star}, \hat{\boldsymbol{\theta}}_n, \mathbf{x}) \right] + 2L\mathbb{E}\left[ \|\hat{\boldsymbol{\theta}}_n - \boldsymbol{\theta}^\star\|_2 \right]$$

$$\leq \mathbb{E}\left[ H(\hat{\mathbf{r}}_n, \hat{\boldsymbol{\theta}}_n, \mathbf{x}) \right] - \frac{1}{n}\sum_{i=1}^{n} H(\hat{\mathbf{r}}_n, \hat{\boldsymbol{\theta}}_n, \mathbf{x}_i) + \frac{1}{n}\sum_{i=1}^{n} H(\mathbf{r}_{f^\star}, \hat{\boldsymbol{\theta}}_n, \mathbf{x}_i)$$

$$- \mathbb{E}\left[ H(\mathbf{r}_{f^\star}, \hat{\boldsymbol{\theta}}_n, \mathbf{x}) \right] + 2L\mathbb{E}\left[ \|\hat{\boldsymbol{\theta}}_n - \boldsymbol{\theta}^\star\|_2 \right]$$

where in the last inequality we used the fact that $\hat{\mathbf{r}}_n$ is a minimizer of $\mathbf{r} \mapsto \frac{1}{n}\sum_{i=1}^{n} H(\mathbf{r}, \hat{\boldsymbol{\theta}}_n, \mathbf{x}_i)$. Finally by using lemmas Lemma 5 and Lemma 6 with $\delta/2$ each, we have that with probability $1 - \delta$,

$$\mathcal{L}_{\boldsymbol{\theta}^\star}(\hat{\mathbf{r}}_n) - \mathcal{L}_{\boldsymbol{\theta}^\star}(\mathbf{r}_{f^\star}) \leq \frac{4}{\sqrt{n}}\mathrm{Rad}(\mathcal{F}) + 2L\mathbb{E}\left[ \|\hat{\boldsymbol{\theta}}_n - \boldsymbol{\theta}^\star\|_2 \right] + 4B\sqrt{\frac{\log(4/\delta)}{n}}$$

Plugging this back into Eq. (E.1), we have that

$$\|\hat{\mathbf{r}}_n - \mathbf{r}_{f^\star}\|_2^2 \leq \frac{2}{\mu p_{\min}} \left( \frac{4}{\sqrt{n}}\mathrm{Rad}(\mathcal{F}) + 4B\sqrt{\frac{\log(4/\delta)}{n}} \right) + \frac{4L}{\mu p_{\min}}\mathbb{E}\left[ \|\hat{\boldsymbol{\theta}}_n - \boldsymbol{\theta}^\star\|_2 \right].$$

■

**Lemma 1.** *For any* $\mathbf{r} \in \mathcal{R}$, $\boldsymbol{\theta} \in \Theta$, $\mathbf{x} \in \mathcal{X}$, *we have that*

$$\mathbf{r}^\top f(\mathbf{x}, \boldsymbol{\theta}) \leq \frac{1}{p_{min}}.$$

*Proof.* Applying Hölder's inequality we have that

$$\mathbf{r}^\top f(\mathbf{x}, \boldsymbol{\theta}) \leq \|\mathbf{r}\|_\infty \|f(\mathbf{x}, \boldsymbol{\theta})\|_1 = \|\mathbf{r}\|_\infty.$$

Moreover, since $\mathbf{r} \in \mathcal{R}$, we have that $\sum_y r_y p_{tr}(y) = 1$ This implies that $\|\mathbf{r}\|_\infty \leq \frac{1}{p_{\min}}$, which yields the result. ∎

**Lemma 2** (Implication of Assumption Assumption 1). *Under Assumption 1, there exists $B > 0$ such that for any* $\mathbf{r} \in \mathcal{R}$, $\boldsymbol{\theta} \in \Theta$, $\mathbf{x} \in \mathcal{X}$,

$$|\log(\mathbf{r}^\top f(\mathbf{x}, \boldsymbol{\theta}))| \leq B.$$

*Proof.* Since $\mathbf{r} \in \mathcal{R}$, it has at least one non-zero coordinate and $f(\mathbf{x}, \boldsymbol{\theta})$ is the output of a softmax layer so all of its coordinates are non-zero. Consequently,

$$\mathbf{r}^\top f(\mathbf{x}, \boldsymbol{\theta}) > 0$$

So by Assumption 1, the function $(\mathbf{r}, \boldsymbol{\theta}, \mathbf{x}) \mapsto \log(\mathbf{r}^\top f(\mathbf{x}, \boldsymbol{\theta}))$ is defined and continuous over a compact set, so there exists a constant $B$ giving us the result. ∎

**Lemma 3** (Population strong convexity). *Let $H(\mathbf{r}, \boldsymbol{\theta}, \mathbf{x}) = -\log(\mathbf{r}^\top f(\mathbf{x}, \boldsymbol{\theta}))$. Under assumption Assumption 2, the function*

$$\mathcal{L}_{\boldsymbol{\theta}^\star} : \mathbf{r} \mapsto \mathbb{E}\left[H(\mathbf{r}, \boldsymbol{\theta}^\star, \mathbf{x})\right]$$

*is $\mu p_{\min}$-strongly convex.*

*Proof.* We first compute the hessian of $\mathcal{L}$ to find that

$$\nabla^2 \mathcal{L}(\mathbf{r}) = \mathbb{E}\left[\frac{1}{(\mathbf{r}^\top f(\mathbf{x}, \boldsymbol{\theta}^\star))^2} f(\mathbf{x}, \boldsymbol{\theta}^\star) f(\mathbf{x}, \boldsymbol{\theta}^\star)^\top\right].$$

Since by Lemma 1, we have that $\mathbf{r}^\top f(\mathbf{x}, \boldsymbol{\theta}^\star) \leq p_{\min}^{-1}$, we have that

$$\nabla^2 \mathcal{L}(\mathbf{r}) \succeq p_{\min} \mathbb{E}\left[f(\mathbf{x}, \boldsymbol{\theta}^\star) f(\mathbf{x}, \boldsymbol{\theta}^\star)^\top\right] \succeq \mu p_{\min} \mathbf{I}_m.$$

∎

**Lemma 4** (Lipschitz parametrization). *Let $H(\mathbf{r}, \boldsymbol{\theta}, \mathbf{x}) = -\log(f(\mathbf{x}, \boldsymbol{\theta})^\top \mathbf{r})$. There exists $L > 0$ such that for any $\boldsymbol{\theta}_1, \boldsymbol{\theta}_2 \in \Theta$, and $\mathbf{r} \in \mathcal{R}$, we have that*

$$|H(\mathbf{r}, \boldsymbol{\theta}_1, \mathbf{x}) - H(\mathbf{r}, \boldsymbol{\theta}_2, \mathbf{x})| \leq L\|\boldsymbol{\theta}_1 - \boldsymbol{\theta}_2\|_2.$$

*Proof.* The gradient of $H$ with respect to $\boldsymbol{\theta}$ is given by

$$\nabla_{\boldsymbol{\theta}} H(\mathbf{r}, \boldsymbol{\theta}, \mathbf{x}) = -\frac{1}{f(\mathbf{x}, \boldsymbol{\theta})^\top \mathbf{r}} \nabla_{\boldsymbol{\theta}} f(\mathbf{x}, \boldsymbol{\theta})$$

Reasoning like in Lemma 1, we know that $\frac{1}{f(\mathbf{x}, \boldsymbol{\theta})^\top \mathbf{r}}$ is defined and continous over the compact set of its parameters, we also know that $f$ is a neural network parametrized by $\boldsymbol{\theta}$, hence $\nabla_{\boldsymbol{\theta}} f(\mathbf{x}, \boldsymbol{\theta})$ is bounded when $\boldsymbol{\theta}$ and $\mathbf{x}$ are bounded. Consequently, under Assumption 1 there exists a constant $L > 0$ such that

$$\|\nabla_{\boldsymbol{\theta}} H(\mathbf{r}, \boldsymbol{\theta}, \mathbf{x})\|_2 \leq L.$$

∎

**Lemma 5** (Uniform bound 1). *Let $\delta \in (0, 1)$, with probability $1 - \delta$, we have that*

$$\mathbb{E}\left[H(\hat{\mathbf{r}}_n, \hat{\boldsymbol{\theta}}_n, \mathbf{x})\right] - \frac{1}{n}\sum_{i=1}^n H(\hat{\mathbf{r}}_n, \hat{\boldsymbol{\theta}}_n, \mathbf{x}_i) \leq \frac{2}{\sqrt{n}}\mathrm{Rad}(\mathcal{F}) + 2B\sqrt{\frac{\log(4/\delta)}{n}}.$$

*Proof.* Let $\delta \in (0,1)$. As $\hat{\mathbf{r}}_n$ is learnt from the samples $\mathbf{x}_i$, we do not have independence which would have allowed us to apply a concentration inequality. Hence, we will derive a uniform bound as follows. We begin by observing that

$$\mathbb{E}\left[H(\hat{\mathbf{r}}_n, \hat{\boldsymbol{\theta}}_n, \mathbf{x})\right] - \frac{1}{n}\sum_{i=1}^{n} H(\hat{\mathbf{r}}_n, \hat{\boldsymbol{\theta}}_n, \mathbf{x}_i) \le \sup_{\mathbf{r}, \boldsymbol{\theta}} \left(\mathbb{E}\left[H(\mathbf{r}, \boldsymbol{\theta}, \mathbf{x})\right] - \frac{1}{n}\sum_{i=1}^{n} H(\mathbf{r}, \boldsymbol{\theta}, \mathbf{x}_i)\right)$$

Now since Lemma 2 holds, we can apply McDiarmid's Inequality to get that with probability $1 - \delta$, we have that

$$\sup_{\mathbf{r}, \boldsymbol{\theta}} \left(\mathbb{E}\left[H(\mathbf{r}, \boldsymbol{\theta}, \mathbf{x})\right] - \frac{1}{n}\sum_{i=1}^{n} H(\mathbf{r}, \boldsymbol{\theta}, \mathbf{x}_i)\right)$$
$$\le \mathbb{E}\left[\sup_{\mathbf{r}, \boldsymbol{\theta}} \left(\mathbb{E}\left[H(\mathbf{r}, \boldsymbol{\theta}, \mathbf{x})\right] - \frac{1}{n}\sum_{i=1}^{n} H(\mathbf{r}, \boldsymbol{\theta}, \mathbf{x}_i)\right)\right] + 2B\sqrt{\frac{\log(2/\delta)}{n}}$$

The expectation of the supremum on the right hand side can be bounded by the Rademacher complexity of $\mathcal{F} := \{\mathbf{x} \mapsto \mathbf{r}^\top f(\mathbf{x}, \boldsymbol{\theta}), \ (\mathbf{r}, \boldsymbol{\theta}) \in \mathcal{R} \times \Theta\}$ and we obtain that

$$\sup_{\mathbf{r}, \boldsymbol{\theta}} \left(\mathbb{E}\left[H(\mathbf{r}, \boldsymbol{\theta}, \mathbf{x})\right] - \frac{1}{n}\sum_{i=1}^{n} H(\mathbf{r}, \boldsymbol{\theta}, \mathbf{x}_i)\right) \le \frac{2}{\sqrt{n}}\mathrm{Rad}(\mathcal{F}) + 2B\sqrt{\frac{\log(2/\delta)}{n}}.$$

∎

**Lemma 6** (Uniform bound 2). *Let $\delta \in (0,1)$, with probability $1 - \delta$, we have that*

$$\mathbb{E}\left[H(\mathbf{r}_{f^\star}, \hat{\boldsymbol{\theta}}_n, \mathbf{x})\right] - \frac{1}{n}\sum_{i=1}^{n} H(\mathbf{r}_{f^\star}, \hat{\boldsymbol{\theta}}_n, \mathbf{x}_i) \le \frac{2}{\sqrt{n}}\mathrm{Rad}(\mathcal{F}) + 2B\sqrt{\frac{\log(2/\delta)}{n}}$$

*Proof.* The proof is identical to that of Lemma 5. ∎

# F   PROOF OF THEOREM 2 AND CONVERGENCE-COMMUNICATION GUARANTEES FOR EQ. (IW-ERM)

We now establish convergence rates for Eq. (IW-ERM) with VRLS and show our proposed importance weighting achieves *the same rates* with the data-dependent *constant terms* increase linearly with $\max_{\mathbf{y} \in \mathcal{Y}} \sup_f r_f(\mathbf{y}) = r_{\max}$ under negligible communication overhead over the baseline ERM-solvers without importance weighting. In Appendix F, we establish tight convergence rates and communication guarantees for Eq. (IW-ERM) with VRLS in a broad range of importance optimization settings including convex optimization, second-order differentiability, composite optimization with proximal operator, optimization with adaptive step-sizes, and nonconvex optimization, along the lines of e.g., (Woodworth et al., 2020; Haddadpour et al., 2021; Glasgow et al., 2022; Liu et al., 2023; Hu and Huang, 2023; Wu et al., 2023; Liu et al., 2023).

By estimating the ratios locally and absorbing into local losses, we note that the properties of the modified local loss w.r.t. the neural network parameters $\mathbf{w}$, *e.g.,* convexity and smoothness, do not change. The data-dependent parameters such as Lipschitz and smoothness constants for $\ell \circ h_{\mathbf{w}}$ w.r.t. $\mathbf{w}$ are scaled linearly by $r_{\max}$. Our method of density ratio estimation trains the pre-defined predictor *exclusively using local training data*, which implies Eq. (IW-ERM) with VRLS achieves the same privacy guarantees as the baseline ERM-solvers without importance weighting. For ratio estimation, the communication between clients involves only the estimated marginal label distribution, instead of data, ensuring negligible communication overhead. Given the size of variables to represent marginal distributions, which is by orders of magnitude smaller than the number of parameters of the underlying neural networks for training and the fact that ratio estimation involves only one round of communication, the overall communication overhead for ratio estimation is masked by the communication costs of model training. The communication costs for  Eq. (IW-ERM) with VRLS

over the course of optimization are exactly the same as those of the baseline ERM-solvers without importance weighting. All in all, importance weighting does not negatively impact communication guarantees throughout the course of optimization, which proves Theorem 2.

In the following, we establish tight convergence rates and communication guarantees for Eq. (IW-ERM) with VRLS in a broad range of importance optimization settings including convex optimization, second-order differentiability, composite optimization with proximal operator, optimization with adaptive step-sizes, and nonconvex optimization.

**Assumption 3** (Convex and Smooth). 1) A minimizer $\mathbf{w}^\star$ exists with bounded $\|\mathbf{w}^\star\|_2$; 2) The $\ell \circ h_\mathbf{w}$ is $\beta$-smoothness and convex w.r.t. $\mathbf{w}$; 3) The stochastic gradient $\mathbf{g}(\mathbf{w}) = \widetilde{\nabla}_\mathbf{w} \ell(h_\mathbf{w})$ is unbiased, i.e., $\mathbb{E}[\mathbf{g}(\mathbf{w})] = \nabla_\mathbf{w} \ell(h_\mathbf{w})$ for any $\mathbf{w} \in \mathcal{W}$ with bounded variance $\mathbb{E}[\|\mathbf{g}(\mathbf{w}) - \nabla_\mathbf{w} \ell(h_\mathbf{w})\|_2^2]$.

For convex and smooth optimization, we establish convergence rates for Eq. (IW-ERM) with VRLS and local updatingz along the lines of e.g., (Woodworth et al., 2020, Theorem 2).

**Theorem 3** (Upper Bound for Convex and Smooth). *Let $D = \|\mathbf{w}_0 - \mathbf{w}^\star\|$, $\tau$ denote the number of local steps (number of stochastic gradients per round of communication per client), $R$ denote the number of communication rounds, and $\max_{\mathbf{y} \in \mathcal{Y}} \sup_f r_f(\mathbf{y}) = r_{\max}$. Under Assumption 3, suppose Algorithm 2 with $\tau$ local updates is run for $T = \tau R$ total stochastic gradients per client with an optimally tuned and constant step-size. Then we have the following upper bound:*

$$\mathbb{E}[\ell(h_{\mathbf{w}_T}) - \ell(h_{\mathbf{w}^\star})] \lesssim \frac{r_{\max}\beta D^2}{\tau R} + \frac{(r_{\max}\beta D^4)^{1/3}}{(\sqrt{\tau}R)^{2/3}} + \frac{D}{\sqrt{K\tau R}}.$$

**Assumption 4** (Convex and Second-order Differentiable). 1) The $\ell(h_\mathbf{w}(\mathbf{x}), \mathbf{y})$ is $\beta$-smoothness and convex w.r.t. $\mathbf{w}$ for any $(\mathbf{x}, \mathbf{y})$; 2) The stochastic gradient $\mathbf{g}(\mathbf{w}) = \widetilde{\nabla}_\mathbf{w} \ell(h_\mathbf{w})$ is unbiased, i.e., $\mathbb{E}[\mathbf{g}(\mathbf{w})] = \nabla_\mathbf{w} \ell(h_\mathbf{w})$ for any $\mathbf{w} \in \mathcal{W}$ with bounded variance $\mathbb{E}[\|\mathbf{g}(\mathbf{w}) - \nabla_\mathbf{w} \ell(h_\mathbf{w})\|_2^2]$.

For convex and second-order Differentiable optimization, we establish a lower bound on the convergence rates for Eq. (IW-ERM) with VRLS and local updating along the lines of e.g., (Glasgow et al., 2022, Theorem 3.1).

**Theorem 4** (Lower Bound for Convex and Second-order Differentiable). *Let $D = \|\mathbf{w}_0 - \mathbf{w}^\star\|$, $\tau$ denote the number of local steps, $R$ denote the number of communication rounds, and $\max_{\mathbf{y} \in \mathcal{Y}} \sup_f r_f(\mathbf{y}) = r_{\max}$. Under Assumption 4, suppose Algorithm 2 with $\tau$ local updates is run for $T = \tau R$ total stochastic gradients per client with a tuned and constant step-size. Then we have the following lower bound:*

$$\mathbb{E}[\ell(h_{\mathbf{w}_T}) - \ell(h_{\mathbf{w}^\star})] \gtrsim \frac{r_{\max}\beta D^2}{\tau R} + \frac{(r_{\max}\beta D^4)^{1/3}}{(\sqrt{\tau}R)^{2/3}} + \frac{D}{\sqrt{K\tau R}}.$$

**Assumption 5** (PL with Compression). 1) The $\ell(h_\mathbf{w}(\mathbf{x}), \mathbf{y})$ is $\beta$-smoothness and convex w.r.t. $\mathbf{w}$ for any $(\mathbf{x}, \mathbf{y})$ and satisfies Polyak-Łojasiewicz (PL) condition (there exists $\alpha_\ell > 0$ such that, for all $\mathbf{w} \in \mathcal{W}$, we have $\ell(h_\mathbf{w}) \leq \|\nabla_\mathbf{w} \ell(h_\mathbf{w})\|_2^2/(2\alpha_\ell)$; 2) The compression scheme $\mathcal{Q}$ is unbiased with bounded variance, i.e., $\mathbb{E}[\mathcal{Q}(\mathbf{x})] = \mathbf{x}$ and $\mathbb{E}[\|\mathcal{Q}(\mathbf{x}) - \mathbf{x}\|_2^2 \leq q\|\mathbf{x}\|_2^2]$; 3) The stochastic gradient $\mathbf{g}(\mathbf{w}) = \widetilde{\nabla}_\mathbf{w} \ell(h_\mathbf{w})$ is unbiased, i.e., $\mathbb{E}[\mathbf{g}(\mathbf{w})] = \nabla_\mathbf{w} \ell(h_\mathbf{w})$ for any $\mathbf{w} \in \mathcal{W}$ with bounded variance $\mathbb{E}[\|\mathbf{g}(\mathbf{w}) - \nabla_\mathbf{w} \ell(h_\mathbf{w})\|_2^2]$.

For nonconvex optimization with PL condition and communication compression, we establish convergence and communication guarantees for Eq. (IW-ERM) with VRLS, compression, and local updating along the lines of e.g., (Haddadpour et al., 2021, Theorem 5.1).

**Theorem 5** (Convergence and Communication Bounds for Nonconvex Optimization with PL). *Let $\kappa$ denote the condition number, $\tau$ denote the number of local steps, $R$ denote the number of communication rounds, and $\max_{\mathbf{y} \in \mathcal{Y}} \sup_f r_f(\mathbf{y}) = r_{\max}$. Under Assumption 5, suppose Algorithm 2 with $\tau$ local updates and communication compression (Haddadpour et al., 2021, Algorithm 1) is run for $T = \tau R$ total stochastic gradients per client with fixed step-sizes $\eta = 1/(2r_{\max}\beta\gamma\tau(q/K + 1))$ and $\gamma \geq K$. Then we have $\mathbb{E}[\ell(h_{\mathbf{w}_T}) - \ell(h_{\mathbf{w}^\star})] \leq \epsilon$ by setting*

$$R \lesssim \left(\frac{q}{K} + 1\right)\kappa \log\left(\frac{1}{\epsilon}\right) \quad \text{and} \quad \tau \lesssim \left(\frac{q + 1}{K(q/K + 1)\epsilon}\right).$$

**Assumption 6** (Nonconvex Optimization with Adaptive Step-sizes). 1) The $\ell \circ h_{\mathbf{w}}$ is $\beta$-smoothness with bounded gradients; 2) The stochastic gradients $\mathbf{g}(\mathbf{w}) = \widetilde{\nabla}_{\mathbf{w}} \ell(h_{\mathbf{w}})$ is unbiased with bounded variance $\mathbb{E}[\|\mathbf{g}(\mathbf{w}) - \nabla_{\mathbf{w}} \ell(h_{\mathbf{w}})\|_2^2]$; 3) Adaptive matrices $A_t$ constructed as in (Wu et al., 2023, Algorithm 2) are diagonal and the minimum eigenvalues satisfy $\lambda_{\min}(A_t) \geq \rho > 0$ for some $\rho \in \mathbb{R}_+$.

For nonconvex optimization with adaptive step-sizes, we establish convergence and communication guarantees for Eq. (IW-ERM) with VRLS and local updating along the lines of e.g., (Wu et al., 2023, Theorem 2).

**Theorem 6** (Convergence and Communication Guarantees for Nonconvex Optimization with Adaptive Step-sizes). *Let $\tau$ denote the number of local steps, $R$ denote the number of communication rounds, and $\max_{\mathbf{y} \in \mathcal{Y}} \sup_f r_f(\mathbf{y}) = r_{\max}$. Under Assumption 6, suppose Algorithm 2 with $\tau$ local updates is run for $T = \tau R$ total stochastic gradients per client with an adaptive step-size similar to (Wu et al., 2023, Algorithm 2). Then we $\mathbb{E}[\|\nabla_{\mathbf{w}} \ell(h_{\mathbf{w}_T})\|_2] \leq \epsilon$ by setting:*

$$T \lesssim \frac{r_{\max}}{K \epsilon^3} \quad \text{and} \quad R \lesssim \frac{r_{\max}}{\epsilon^2}.$$

**Assumption 7** (Composite Optimization with Proximal Operator). 1) The $\ell \circ h_{\mathbf{w}}$ is smooth and strongly convex with condition number $\kappa$; 2) The stochastic gradients $\mathbf{g}(\mathbf{w}) = \widetilde{\nabla}_{\mathbf{w}} \ell(h_{\mathbf{w}})$ is unbiased.

For composite optimization with strongly convex and smooth functions and proximal operator, we establish an upper bound on oracle complexity to achieve $\epsilon$ error on the Lyapunov function defined as in (Hu and Huang, 2023, Section 4) for Gradient Flow-type transformation of Eq. (IW-ERM) with VRLS in the limit of infinitesimal step-size.

**Theorem 7** (Oracle Complexity of Proximal Operator for Composite Optimization). *Let $\kappa$ denote the condition number. Under Assumption 7, suppose Gradient Flow-type transformation of Algorithm 2 with VRLS and Proximal Operator evolves in the limit of infinitesimal step-size (Hu and Huang, 2023, Algorithm 3). Then it achieves $\mathcal{O}\left(r_{\max} \sqrt{\kappa} \log(1/\epsilon)\right)$ Proximal Operator Complexity.*

We finally establish high-probability convergence bounds for Eq. (IW-ERM) with VRLS along the lines of e.g., (Liu et al., 2023, Theorem 4.1). To show the impact of importance weighting on convergence rate decoupled from the impact of number of clients and obtain the current SotA *high-probability* bounds for nonconvex optimization, we focus on Eq. (IW-ERM) with $K = 1$.

We assume the following mild assumptions in *nonconvex* optimization (Liu et al., 2023).

**Assumption 8** (Sub-Gaussian Noise). 1) A minimizer $\mathbf{w}^\star$ exists; 2) The stochastic gradients $\mathbf{g}(\mathbf{w}) = \widetilde{\nabla}_{\mathbf{w}} \ell(h_{\mathbf{w}})$ is unbiased, i.e., $\mathbb{E}[\mathbf{g}(\mathbf{w})] = \nabla_{\mathbf{w}} \ell(h_{\mathbf{w}})$ for any $\mathbf{w} \in \mathcal{W}$; 3) The noise $\|\mathbf{g}(\mathbf{w}) - \nabla_{\mathbf{w}} \ell(h_{\mathbf{w}})\|_2$ is $\sigma$-sub-Gaussian (Vershynin, 2018).

**Theorem 8** (High-probability Bound for Nonconvex Optimization). *Let $\delta \in (0, 1)$ and $T \in \mathbb{Z}_+$. Let $K = 1$ and $\max_{\mathbf{y} \in \mathcal{Y}} \sup_f r_f(\mathbf{y}) = r_{\max}$. Under Assumption 8 and $\beta$-smoothness of nonconvex $\ell \circ h_{\mathbf{w}}$, suppose Algorithm 2 is run for $T$ iterations with a step-size $\min\left\{\frac{1}{r_{\max}\beta}, \sqrt{\frac{1}{\sigma^2 r_{\max} \beta T}}\right\}$. Then with probability $1 - \delta$, gradient norm squareds satisfy:*

$$\frac{1}{T} \sum_{t=1}^{T} \|\nabla_{\mathbf{w}} \ell(h_{\mathbf{w}_t})\|_2^2 = O\left(\sigma \sqrt{\frac{r_{\max} \beta}{T}} + \frac{\sigma^2 \log(1/\delta)}{T}\right).$$

Theorem 8 shows that when the stochastic gradients are too noisy $\sigma = \Omega(\sqrt{r_{\max}\beta}/\log(1/\delta))$ such that the second term in the rate dominates, then importance weighting does not have any negative impact on the convergence rate.

*Proof.* We note that density ratios do not depend on the model parameters $\mathbf{w}$ and the Lipschitz and smoothness constants for $\ell \circ h_{\mathbf{w}}$ w.r.t. $\mathbf{w}$ are scaled by $r_{\max}$. The rest of the proof follows the arguments of (Liu et al., 2023, Theorem 4.1). ∎

## G   COMPARISON OF LATENT DISTRIBUTION

**Dirac Delta on Input**:

$$p(\mathbf{z}|\mathbf{x}) = \delta_{\mathbf{x}} \tag{G.1}$$

This definition implies that the latent variable $\mathbf{z}$ is directly equal to the input $\mathbf{x}$, i.e $\mathbf{z} = \mathbf{x}$ in MLLS. Here, the distribution in this latent space is a Dirac delta function centered at data point $\mathbf{x}$.

**Dirac Delta on Predictor Output**:

$$p(\mathbf{z}|\mathbf{x}) = \delta_{\arg\max f(\mathbf{x})} \tag{G.2}$$

In this scenario, $\mathbf{z}$ is essentially the class (or dimension) with the maximum output value from $f(\mathbf{x})$. Conceptually, as BBSE proposed, $\mathbf{z}$ represents the predicted class, and its distribution is a Dirac delta function centered on the class with the highest probability.

**Kernel-Transformed Predictor Output**:

$$p(\mathbf{z}|\mathbf{x}) = g(f(\mathbf{x})) \tag{G.3}$$

Here, $\mathbf{z}$ represents a distribution in a latent space, which is transformed from the outputs $f(\mathbf{x})$ by the kernel function $g$. The specific nature of this distribution is contingent on the kernel $g$. For example, when $g$ is an RBF kernel, the latent space possesses a Gaussian distribution centered at the values of $f(\mathbf{x})$. This can be conceptualized as a generalized version of a Dirac delta function. Intriguingly, with a linear kernel, no prior distribution is defined on the output of $f(\mathbf{x})$. Nevertheless, the linear kernel is still utilized by us for similarity comparisons on MNIST and Fashion MNIST datasets.

These representations offer a range of choices for modeling the latent space, each with its own characteristics. The choice between them determines different definitions of calibration methods.

In addition to the MMD, by employing the Kantorovich-Rubinstein Duality (Dedecker et al., 2006), analogous to the approach used in Wasserstein GANs (Arjovsky et al., 2017), we can define a Wasserstein Distance as:

$$\mathrm{W}(\mathbf{z}_1, \mathbf{z}_2 \mid \mathbf{x}, \alpha) = \sup_{\|g\| \leq 1} \left( \mathbb{E}_{q(\mathbf{z}_1|\mathbf{x},\alpha)}[g(f(\mathbf{x}))] - \mathbb{E}_{\mathbf{z}_2|\mathbf{x},\alpha}[g(f(\mathbf{x}))] \right) \tag{G.4}$$

where $g : \mathbb{R}^m \to \mathbb{R}$ is a learnable function satisfying the 1-Lipschitz constraint. Practically, the $\sup$ operation is approximated by $\arg\max$. Minimizing this distance entails a min-max game: given a fixed $f$, we aim to maximize the discrepancy with respect to $g$, and with a fixed $g$, we seek to minimize the discrepancy with respect to $f$.

## H   EXPERIMENTAL DETAILS AND ADDITIONAL EXPERIMENTS

In this section, we provide experimental details and additional experiments. In particular, we validate our theory on multiple clients in a federated setting and show that our IW-ERM outperforms FedAvg and FedBN baselines i *under drastic and challenging label shifts*.

### H.1   EXPERIMENTAL DETAILS

In single-client experiments, a simple MLP without dropout is used as the predictor for MNIST, and ResNet-18 for CIFAR-10.

For experiments in a federated learning setting, both MNIST (LeCun et al., 1998) and Fashion MNIST (Xiao et al., 2017) datasets are employed, each containing 60,000 training samples and 10,000 test samples, with each sample being a 28 by 28 pixel grayscale image. The CIFAR-10 dataset (Krizhevsky) comprises 60,000 colored images, sized 32 by 32 pixels, spread across 10 classes with 6,000 images per class; it is divided into 50,000 training images and 10,000 test images. In this setting, the objective is to minimize the cross-entropy loss. Stochastic gradients for each client are calculated with a batch size of 64 and aggregated on the server using the Adam optimizer. LeNet is used for experiments on MNIST and Fashion MNIST with a learning rate of 0.001 and a

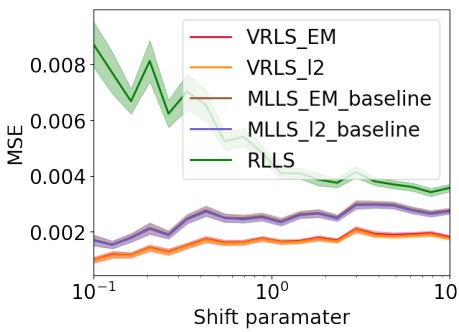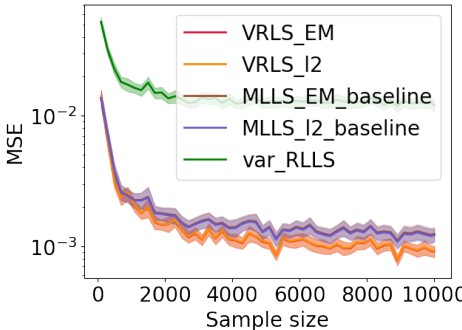

**Figure 6:** *In this experiment with MNIST, we compare VRLS with MLLS, EM and RLLS.*

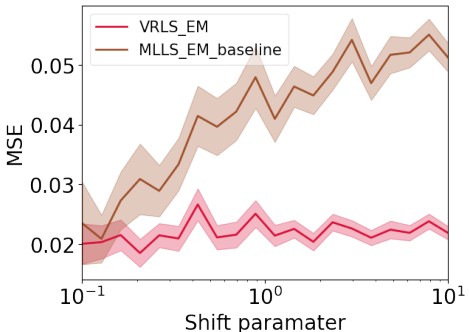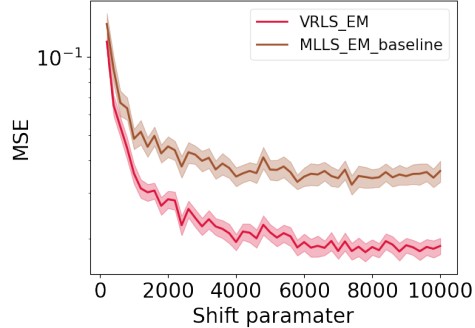

**Figure 7:** *In this experiment with Fashion MNIST, as menthoned previously, a linear kernel and a simple MLP with dropout were employed. For the sake of privacy in Federated Learning (FL) applications, we selectively chose a setting wherein our EM method exhibited a low MSE, and thus, only results from the EM method are depicted. It's important to clarify that the main objective of this illustration is not a comparative analysis between MLLS and VRLS, hence a configuration yielding lower MSE loss for VRLS_EM was adopted.*

weight decay of $1 \times 10^{-6}$. For CIFAR-10, ResNet-18 is employed with a learning rate of 0.0001 and a weight decay of 0.0001. Three independent runs are implemented for 5-client experiments on Fashion MNIST and CIFAR-10, while for 10 clients, one run is conducted on CIFAR-10. All experiments are performed using a single GPU on an internal cluster and Colab.

Importantly, the training of the predictor for ratio estimation on both the baseline MLLS and our VRLS is executed with identical hyperparameters and epochs for CIFAR-10 and Fashion MNIST. The training is halted once the classification loss reaches a predefined threshold on MNIST.

## H.2 Additional experiments

In this section, we provide supplementary results, visualizations of accuracy across clients and tables showing dataset distribution in FL setting.

## I Complexity analysis

In our algorithm, the ratio estimation is executed only once prior to IW-ERM in parallel.

In experiments, we used simple network to estimate the ratios in advance, whose training is not computationally extensive compared to training the global model. Compared to baseline FedAvg, the additional computational complexity of our IW-ERM with VRLS leads to substantial improvements of the overall generalization in settings under challenging label shifts.

**Table 5:** LeNet on Fashion MNIST with label shift across 5 clients. 15000 iterations for FedAvg and FedBN; 5000 for Upper Bound (FTW-ERM) using true ratios and our IW-ERM. To mention, to train our predictor, we use a simpliest MLP and employ linear kernel.

| FMNIST | Our IW-ERM | FedAvg | FedBN | Upper Bound |
|---|---|---|---|---|
| **Avg. accuracy** | **0.7520 $\pm$ 0.0209** | 0.5472 $\pm$ 0.0297 | 0.5359 $\pm$ 0.0306 | 0.8273 $\pm$ 0.0041 |
| Client 1 accuracy | **0.7162 $\pm$ 0.0059** | 0.3616 $\pm$ 0.0527 | 0.3261 $\pm$ 0.0296 | 0.8590 $\pm$ 0.0062 |
| Client 2 accuracy | **0.9266 $\pm$ 0.0125** | 0.9060 $\pm$ 0.0157 | 0.9035 $\pm$ 0.0162 | 0.9357 $\pm$ 0.0037 |
| Client 3 accuracy | **0.6724 $\pm$ 0.0467** | 0.3279 $\pm$ 0.0353 | 0.3612 $\pm$ 0.0814 | 0.7896 $\pm$ 0.0109 |
| Client 4 accuracy | **0.7979 $\pm$ 0.0448** | 0.6858 $\pm$ 0.0105 | 0.6654 $\pm$ 0.0121 | 0.8098 $\pm$ 0.0112 |
| Client 5 accuracy | **0.6468 $\pm$ 0.0248** | 0.4548 $\pm$ 0.0655 | 0.4234 $\pm$ 0.0387 | 0.7426 $\pm$ 0.0257 |

**Table 6:** ResNet-18 on CIFAR-10 with label shift across 5 clients. For fair comparison, we run 5000 iterations for our method and Upper Bound, while 10000 for FedAvg and FedBN.

| CIFAR-10 | Our IW-ERM | FedAvg | FedBN | Upper Bound |
|---|---|---|---|---|
| **Avg. accuracy** | **0.5640 $\pm$ 0.0241** | 0.4515 $\pm$ 0.0148 | 0.4263 $\pm$ 0.0975 | 0.5790 $\pm$ 0.0103 |
| Client 1 accuracy | **0.6410 $\pm$ 0.0924** | 0.5405 $\pm$ 0.1845 | 0.5321 $\pm$ 0.0620 | 0.7462 $\pm$ 0.0339 |
| Client 2 accuracy | **0.8434 $\pm$ 0.0359** | 0.3753 $\pm$ 0.0828 | 0.4656 $\pm$ 0.2158 | 0.7509 $\pm$ 0.0534 |
| Client 3 accuracy | **0.4591 $\pm$ 0.1131** | 0.3973 $\pm$ 0.1333 | 0.2838 $\pm$ 0.1055 | 0.5845 $\pm$ 0.0854 |
| Client 4 accuracy | **0.4751 $\pm$ 0.1241** | 0.5007 $\pm$ 0.1303 | 0.5256 $\pm$ 0.1932 | 0.3507 $\pm$ 0.0578 |
| Client 5 accuracy | **0.4013 $\pm$ 0.0430** | 0.4429 $\pm$ 0.1195 | 0.5603 $\pm$ 0.1581 | 0.4627 $\pm$ 0.0456 |

**Table 7:** Label distribution on Fasion MNIST with 5 clients, with the majority of classes possessing a limited number of training and test images across each client.

| | | Class | | | | | | | | | |
|---|---|---|---|---|---|---|---|---|---|---|---|
| | | 0 | 1 | 2 | 3 | 4 | 5 | 6 | 7 | 8 | 9 |
| Client 1 | Train | 34 | 34 | 34 | 34 | 34 | 5862 | 34 | 34 | 34 | 34 |
| | Test | 977 | 5 | 5 | 5 | 5 | 5 | 5 | 5 | 5 | 5 |
| Client 2 | Train | 34 | 34 | 34 | 34 | 34 | 34 | 5862 | 34 | 34 | 34 |
| | Test | 5 | 977 | 5 | 5 | 5 | 5 | 5 | 5 | 5 | 5 |
| Client 3 | Train | 34 | 34 | 34 | 34 | 34 | 34 | 34 | 5862 | 34 | 34 |
| | Test | 5 | 5 | 977 | 5 | 5 | 5 | 5 | 5 | 5 | 5 |
| Client 4 | Train | 34 | 34 | 34 | 34 | 34 | 34 | 34 | 34 | 5862 | 34 |
| | Test | 5 | 5 | 5 | 977 | 5 | 5 | 5 | 5 | 5 | 5 |
| Client 5 | Train | 34 | 34 | 34 | 34 | 34 | 34 | 34 | 34 | 34 | 5862 |
| | Test | 5 | 5 | 5 | 5 | 977 | 5 | 5 | 5 | 5 | 5 |

## J  MATHEMATICAL NOTATIONS

In this appendix, we provide a summary of mathematical notations used in this paper in Table 10:

## K  LIMITATIONS

- The merits of our approach are not entirely elucidated as the maximization over variational distributions is non-differentiable. Rather, we explicitly define a worst-case scenario and accordingly select the Dirichlet parameter from a Gaussian distribution. Choosing alternative forms of distributions, such as long-tail distributions primarily skewed towards the negative direction, could more aptly address the "worst-case" scenario. However, for the purposes of this study, we have opted to employ the simplest case.

- As a proxy of real-world settings, we artificially implemented a mild test data augmentation to satisfy the relaxed label shift assumptions and control ratio estimation errors for both baselines and our method. It is an interesting future problem to develop variants of VRLS to

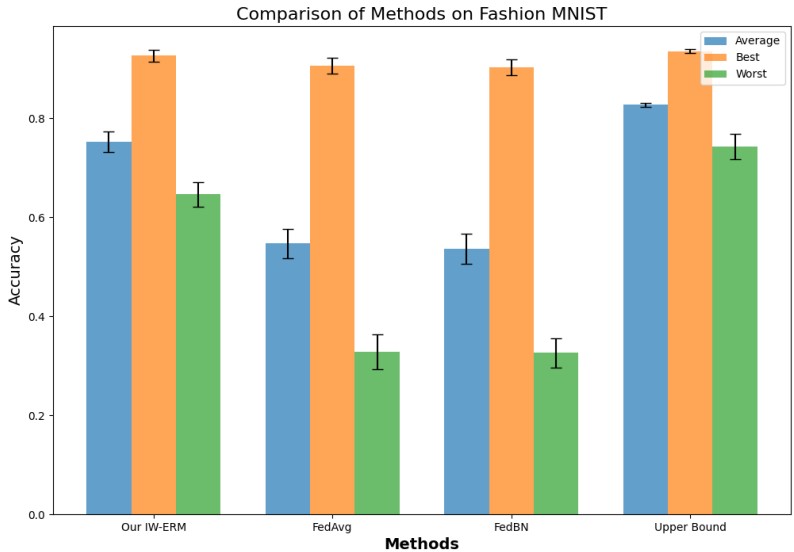

**Figure 8:** *The average, best-client, and worst-client accuracy, along with their standard deviations, are derived from Table 5. Our method exhibits the lowest standard deviation, showcasing the most robust accuracy amongst the compared methods.*

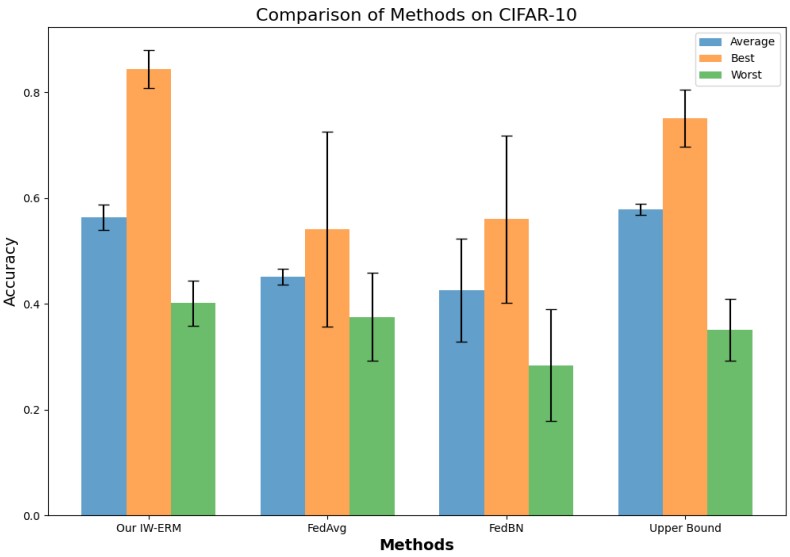

**Figure 9:** *The average, best-client, and worst-client accuracy, along with their standard deviations, are derived from Table 6.*

provably handle general distribution shifts in more challenging settings such as CIFAR-10.1 dataset (Recht et al., 2018; Torralba et al., 2008) as discused in (Garg et al., 2023).

- In the federated learning experiments conducted on CIFAR-10, as similarly reported in (Ramezani-Kebrya et al., 2023), the presence of high standard deviation renders the results less reliable. Tuning the hyper-parameters carefully for training can facilitate more robust global training and thereby enhance the reliability of the results. Additionally, to maintain a

**Table 8:** Label distribution on CIFAR-10 with 5 clients, with the majority of classes possessing a limited number of training and test images across each client.

| | | Class | | | | |
|---|---|---|---|---|---|---|
| | | 0 | 1 | 2 | 3 | 4 |
| Client 1-10 | Train | $^{95}/_{100}$ | $^{5}/_{9}$ | $^{5}/_{9}$ | $^{5}/_{9}$ | $^{5}/_{9}$ |
| | Test | $^{5}/_{9}$ | $^{5}/_{9}$ | $^{5}/_{9}$ | $^{5}/_{9}$ | $^{5}/_{9}$ |
| Client 11-20 | Train | $^{5}/_{9}$ | $^{95}/_{100}$ | $^{5}/_{9}$ | $^{5}/_{9}$ | $^{5}/_{9}$ |
| | Test | $^{5}/_{9}$ | $^{5}/_{9}$ | $^{5}/_{9}$ | $^{5}/_{9}$ | $^{5}/_{9}$ |
| Client 21-30 | Train | $^{5}/_{9}$ | $^{5}/_{9}$ | $^{95}/_{100}$ | $^{5}/_{9}$ | $^{5}/_{9}$ |
| | Test | $^{5}/_{9}$ | $^{5}/_{9}$ | $^{5}/_{9}$ | $^{5}/_{9}$ | $^{5}/_{9}$ |
| Client 31-40 | Train | $^{5}/_{9}$ | $^{5}/_{9}$ | $^{5}/_{9}$ | $^{95}/_{100}$ | $^{5}/_{9}$ |
| | Test | $^{5}/_{9}$ | $^{5}/_{9}$ | $^{5}/_{9}$ | $^{5}/_{9}$ | $^{5}/_{9}$ |
| Client 41-50 | Train | $^{5}/_{9}$ | $^{5}/_{9}$ | $^{5}/_{9}$ | $^{5}/_{9}$ | $^{95}/_{100}$ |
| | Test | $^{5}/_{9}$ | $^{5}/_{9}$ | $^{5}/_{9}$ | $^{5}/_{9}$ | $^{5}/_{9}$ |
| Client 51-60 | Train | $^{5}/_{9}$ | $^{5}/_{9}$ | $^{5}/_{9}$ | $^{5}/_{9}$ | $^{5}/_{9}$ |
| | Test | $^{5}/_{9}$ | $^{5}/_{9}$ | $^{5}/_{9}$ | $^{5}/_{9}$ | $^{95}/_{100}$ |
| Client 61-70 | Train | $^{5}/_{9}$ | $^{5}/_{9}$ | $^{5}/_{9}$ | $^{5}/_{9}$ | $^{5}/_{9}$ |
| | Test | $^{5}/_{9}$ | $^{5}/_{9}$ | $^{5}/_{9}$ | $^{95}/_{100}$ | $^{5}/_{9}$ |
| Client 71-80 | Train | $^{5}/_{9}$ | $^{5}/_{9}$ | $^{5}/_{9}$ | $^{5}/_{9}$ | $^{5}/_{9}$ |
| | Test | $^{5}/_{9}$ | $^{5}/_{9}$ | $^{95}/_{100}$ | $^{5}/_{9}$ | $^{5}/_{9}$ |
| Client 81-90 | Train | $^{5}/_{9}$ | $^{5}/_{9}$ | $^{5}/_{9}$ | $^{5}/_{9}$ | $^{5}/_{9}$ |
| | Test | $^{5}/_{9}$ | $^{95}/_{100}$ | $^{5}/_{9}$ | $^{5}/_{9}$ | $^{5}/_{9}$ |
| Client 91-100 | Train | $^{5}/_{9}$ | $^{5}/_{9}$ | $^{5}/_{9}$ | $^{5}/_{9}$ | $^{5}/_{9}$ |
| | Test | $^{95}/_{100}$ | $^{5}/_{9}$ | $^{5}/_{9}$ | $^{5}/_{9}$ | $^{5}/_{9}$ |

| | | Class | | | | |
|---|---|---|---|---|---|---|
| | | 5 | 6 | 7 | 8 | 9 |
| Client 1-10 | Train | $^{5}/_{9}$ | $^{5}/_{9}$ | $^{5}/_{9}$ | $^{5}/_{9}$ | $^{5}/_{9}$ |
| | Test | $^{5}/_{9}$ | $^{5}/_{9}$ | $^{5}/_{9}$ | $^{5}/_{9}$ | $^{95}/_{100}$ |
| Client 11-20 | Train | $^{5}/_{9}$ | $^{5}/_{9}$ | $^{5}/_{9}$ | $^{5}/_{9}$ | $^{5}/_{9}$ |
| | Test | $^{5}/_{9}$ | $^{5}/_{9}$ | $^{5}/_{9}$ | $^{95}/_{100}$ | $^{5}/_{9}$ |
| Client 21-30 | Train | $^{5}/_{9}$ | $^{5}/_{9}$ | $^{5}/_{9}$ | $^{5}/_{9}$ | $^{5}/_{9}$ |
| | Test | $^{5}/_{9}$ | $^{5}/_{9}$ | $^{95}/_{100}$ | $^{5}/_{9}$ | $^{5}/_{9}$ |
| Client 31-40 | Train | $^{5}/_{9}$ | $^{5}/_{9}$ | $^{5}/_{9}$ | $^{5}/_{9}$ | $^{5}/_{9}$ |
| | Test | $^{5}/_{9}$ | $^{95}/_{100}$ | $^{5}/_{9}$ | $^{5}/_{9}$ | $^{5}/_{9}$ |
| Client 41-50 | Train | $^{5}/_{9}$ | $^{5}/_{9}$ | $^{5}/_{9}$ | $^{5}/_{9}$ | $^{5}/_{9}$ |
| | Test | $^{95}/_{100}$ | $^{5}/_{9}$ | $^{5}/_{9}$ | $^{5}/_{9}$ | $^{5}/_{9}$ |
| Client 51-60 | Train | $^{95}/_{100}$ | $^{5}/_{9}$ | $^{5}/_{9}$ | $^{5}/_{9}$ | $^{5}/_{9}$ |
| | Test | $^{5}/_{9}$ | $^{5}/_{9}$ | $^{5}/_{9}$ | $^{5}/_{9}$ | $^{5}/_{9}$ |
| Client 61-70 | Train | $^{5}/_{9}$ | $^{95}/_{100}$ | $^{5}/_{9}$ | $^{5}/_{9}$ | $^{5}/_{9}$ |
| | Test | $^{5}/_{9}$ | $^{5}/_{9}$ | $^{5}/_{9}$ | $^{5}/_{9}$ | $^{5}/_{9}$ |
| Client 71-80 | Train | $^{5}/_{9}$ | $^{5}/_{9}$ | $^{95}/_{100}$ | $^{5}/_{9}$ | $^{5}/_{9}$ |
| | Test | $^{5}/_{9}$ | $^{5}/_{9}$ | $^{5}/_{9}$ | $^{5}/_{9}$ | $^{5}/_{9}$ |
| Client 81-90 | Train | $^{5}/_{9}$ | $^{5}/_{9}$ | $^{5}/_{9}$ | $^{95}/_{100}$ | $^{5}/_{9}$ |
| | Test | $^{5}/_{9}$ | $^{5}/_{9}$ | $^{5}/_{9}$ | $^{5}/_{9}$ | $^{5}/_{9}$ |
| Client 91-100 | Train | $^{5}/_{9}$ | $^{5}/_{9}$ | $^{5}/_{9}$ | $^{5}/_{9}$ | $^{95}/_{100}$ |
| | Test | $^{5}/_{9}$ | $^{5}/_{9}$ | $^{5}/_{9}$ | $^{5}/_{9}$ | $^{5}/_{9}$ |

**Table 9:** Label distribution on CIFAR-10 with 100 clients, wherein groups of 10 clients share the same distribution and ratios. The majority of classes possess a limited quantity of training and test images on each client.

fair comparison, we did not allow additional training iterations for FedBN, despite its slower convergence.

**Table 10:** Math Symbols

| Math Symbol | Definition |
|---|---|
| $\mathcal{X}$ | Compact metric space for features |
| $\mathcal{Y}$ | Discrete label space with $|\mathcal{Y}| = m$ |
| $K$ | Number of clients in an FL setting |
| $\mathcal{S}_k$ | All samples in the training set of client $k$ |
| $h_{\mathbf{w}}$ | Hypothesis function $h_{\mathbf{w}} : \mathcal{X} \to \mathcal{Y}$ |
| $\mathcal{H}$ | Hypothesis class for $h_{\mathbf{w}}$ |
| $\mathcal{Z}$ | Mapping space from $\mathcal{X}$, which can be discrete or continuous |

- Within the FL framework, VRLS has the potential to require absolutely no data sharing, including the training data. This can be accomplished through a distributed training of the predictor, as illustrated in Algorithm 2. The training data from each client can be regarded as a mini-batch. However, a significant label shift across individual clients can adversely affect the ratio estimation accuracy. Future work can explore a balance between accurate ratio estimation and preserving privacy.

