# OpenReview forum: "Federated Learning under Label Shifts with Guarantees"
_ICLR.cc/2024/Conference — Submitted to ICLR 2024_

### Official Review · Reviewer_VBZD · 2023-11-01

**Soundness:** 2 fair
**Presentation:** 2 fair
**Contribution:** 2 fair
**Rating:** 5
**Confidence:** 3

**Summary:**

This paper studies the label shift adaptation problem under the federated learning setting. There are two parts. For the first part, this paper proposes VRLS, a regularized version of the MLLS method. The authors further show that the optimization problem can be approximately solved by an EM procedure. The second part introduced how to apply the VRLS algorithm to the federated learning setting. Theoretical guarantees on the sample complexity of the density ratio label density ratio estimator and the convergence rate of the IW-ERM algorithm are provided. Extensive experiments are also conducted to evaluate the proposed methods.

**Strengths:**

The strengths of this paper as as follows:
- Interesting problem formulation: this paper studies the label shift problem under the Federated learning setting. The problem formulation is both interesting and practically relevant.
- Superior empirical performance: this paper proposes a regularized version of the MLLS method. Experiments show that the proposed method achieves superior empirical performance than MLLS method.

**Weaknesses:**

The weaknesses of the paper are as follows:
- Unclear main focus: While the paper is titled "Federated Learning under Label Shifts with Guarantees," I noticed that a significant portion appears to study the classical label shift problem in supervised learning. In particular, Section 3 only briefly touches upon applying the proposed density ratio estimation to federated learning, but it lacks a detailed algorithm description and a thorough discussion of its contributions. In Section 4, Theorem 1 and Theorem 2 seem to offer limited insight into how the proposed algorithm might effectively tackle the challenges of federated learning. Specifically, Theorem 1 seems to pertain only to a single client, and the presentation of Theorem 2 is somewhat confusing, as discussed in the next point.

- Clarity of Theorem 2: The statement of Theorem 2 strikes me as somewhat informal, particularly due to my uncertainty regarding the definition of the notation $\ell$. In Section 2, it is defined as the loss function on a single sample. In light of this, it's not unclear to me how the theorem relates to the objective Eq.(IW-ERM) that the authors are trying to minimize. Moreover, the proof of Theorem 2 seems to be a straightforward application of (Liu et al., 2023, Theorem 4.1). It's not clear how this theorem perspective helps to enhance our understanding of the federated learning problem.

- Unclear theoretical advantages: As indicated by Eq (3.1) and Eq. (3.2), the proposed method appears similar to MLLS with the difference that the model $\theta^*$, is trained with an additional regularization term. While the experiments demonstrate the benefit of this additional regularization, Theorem 3 shows a similar convergence rate for the proposed VRLS when compared to MLLS. I think the theorem will be more appealing if the authors can provide a more precise explanation of why the regularization helps with the label shift problem.

- Empirical comparison: a closely related work is [1] as cited by the authors, which also considers adding a regularization term to perform for the label density ratio estimation. I think the experiments will be more convincing by also taking [1] as a compared method.

[1]  Azizzadenesheli et al.  Regularized learning for domain adaptation under label shifts. In ICML 2019.

**Questions:**

1. The connection between the optimization problem in Eq.(Reg-Est) and that shown by Eq.(3.1) and Eq.(3.2) is not immediately clear to me. In Eq.(Reg-Est), the regularization term is incorporated into the training of the density ratio, whereas in Eq.(3.1), it appears to be a part of the loss function to train the classifier $\theta^*$. It would be beneficial if the author could clarify this by providing a more formal statement that establishes the equivalence between these two optimization problems.

2. Could you provide a more comprehensive theoretical explanation of how the proposed help to minimize goal Eq.(IW_ERM) mentioned in Section 2?  (Please refer to the second point of weaknesses for more details.)

3. I think this paper would be more appealing if the authors could show the advantage of the proposed method over MLLS from a theoretical view. (Please refer to the third point of weaknesses for more details.)

---

> ### Author Response · Authors · 2023-11-17
> **Response to Reviewer VBZD**
>
> We thank the reviewer for their thoughtful comments which we address one by one below.
>
> **Empirical comparison**
>
>  For your reference, we upload two plots anonymously [here](https://imgur.com/a/bCFhOTi). Regularized Learning under Label Shift (RLLS) [1] is a variation of BBSE, based on the confusion matrix approach. This method has been already compared with MLLS in [2]. I also implemented this method and obtained consistent results as Figure 1 (b)(e) in [2]. To sum up, RLLS, which is based on the confusion matrix, improved based on BBSE, underperforms divergence-based and EM-based methods. The potential reason has been studied by [2]: “This concludes MLLS’s superior performance is not because of differences in loss function used for distribution matching but due to differences in the granularity of the predictions, caused by crude confusion matrix aggregation”, and “MLLS’s superior performance (when applied with more granular calibration techniques) is not due to its objective but rather to the information lost by BBSE via confusion matrix calibration”.
>
>  **The advantage of the proposed method over MLLS from a theoretical view**
> The generalization bounds in Theorem 1 are there to  highlight the dependence on the complexity of the function class in which the optimization is conducted. The presence of an additional regularizer in our method induces a reduction in the size of the model. We have not been able to characterize exactly this reduction yet, but we believe this reduced complexity is behind our improved results. It is an interesting direction for future work.
>
> **Formal statement that establishes the equivalence between these two optimization problems**
>  As you correctly pointed out, the IW-ERM framework under federated learning can indeed be seen as an extension of the single-client IW-ERM to a multi-client scenario. This generalization is succinctly encapsulated in the optimization problem outlined in section C.3 of the Appendix.
>
> From an optimization perspective, the IW-ERM's adaptation to multiple clients translates to the aggregation of the numerator of the weight ratio for each individual client. This modification not only preserves the essence of the IW-ERM approach but also effectively scales it for federated learning environments.
>
> Regarding the convergence rate, the established bounds, and communication costs, our study presents several theorems that address these aspects in depth. These theorems provide a solid theoretical foundation, demonstrating how the IW-ERM under federated learning efficiently navigates these challenges. Specifically, they illustrate that despite the inherent complexities of federated learning, our approach remains both robust and scalable, effectively managing the balance between communication efficiency and convergence rate.  Additionally, the updated PDF includes various theorems concerning the lower and upper bounds of convergence rate and communication guarantees. Here is a summary: To sum up, we have:
>
> 1. Negligible impact on Convergence Rate and Communication Guarantees (Theorem 2):  $O(r_{max} h(T))$.
> 2. Upper Bound on Convergence Rate for Convex and Smooth Optimization (Theorem 3):
> \begin{align}
> E[l(h_{w_T})-l(h_{w^\star})]
> \lesssim \frac{r_{max}b D^2}{\tau R}+\frac{(r_{max}b D^4)^{1/3}}{(\sqrt{\tau}R)^{2/3}}+\frac{D}{\sqrt{K\tau R}}.
> \end{align}
> 3. Lower Bound on Convergence Rate for for Convex and Second-order Differentiable Optimization (Theorem 4):
> \begin{align}
> E[l(h_{w_T})-l(h_{w^\star})]
> \gtrsim \frac{r_{\max}b D^2}{\tau R}+\frac{(r_{max}b D^4)^{1/3}}{(\sqrt{\tau}R)^{2/3}}+\frac{D}{\sqrt{K\tau R}}.
> \end{align}
> 4. Convergence and Communication Bounds for Nonconvex Optimization for  PL with Compression (Theorem 5):
> \begin{align}
> R\lesssim \Big(\frac{q}{K}+1\Big)\kappa\log\Big(\frac{1}{\epsilon}\Big)\quad and \quad \tau\lesssim\Big(\frac{q+1}{K(q/K+1)\epsilon}\Big).
> \end{align}
> 5. Convergence and Communication Guarantees for Nonconvex Optimization with Adaptive Step-sizes for Non-convex Optimization with Adaptive Step-sizes Assumptions (Theorem 6):
> \begin{align}
> T\lesssim  \frac{r_{max}}{K\epsilon^3}\quad and \quad R\lesssim\frac{r_{max}}{\epsilon^2}.
> \end{align}
> 6. Oracle Complexity of Proximal Operator for  Composite Optimization with Proximal Operator (Theorem 7):  $O\big(r_{max}\sqrt{\kappa}\log(1/\epsilon)\big)$
>
> 7. High-probability Bound for Nonconvex Optimization for Sub-Gaussian Assumption (Theorem 8):
>
> \begin{align}
> \frac{1}{T}\sum_{t=1}^T\|\nabla_{w}l(h_{w_t})\|_2^2= O(\sigma \sqrt{\frac{r_m \beta}{T}} + \frac{\sigma^2\log(1/\delta)}{T}).
> \end{align}
>
>
> [1]. Azizzadenesheli et al. Regularized learning for domain adaptation under label shifts. In ICML 2019.
>
> [2]. Saurabh Garg, Yifan Wu, Sivaraman Balakrishnan, and Zachary C. Lipton. A unified view of label
> shift estimation. In Advances in neural information processing systems (NeurIPS), 2020.

---

> ### Author Response · Authors · 2023-11-20
> **Response to Reviewer VBZD**
>
> Dear Reviewer VBZD,
>
> We would like to express our gratitude for your review of our ICLR submission. We have taken your insightful comments into account and have made the necessary adjustments. Should you have any questions or require further information, please feel free to contact us. We look forward to any additional feedback you may have.

---

> ### Author Response · Authors · 2023-11-23
> **Response to Reviewer VBZD**
>
> This is a friendly reminder that we submitted our responses to your comments and are looking forward to your feedback for further refinement of our paper.

---

### Official Review · Reviewer_PGzm · 2023-11-02

**Soundness:** 3 good
**Presentation:** 2 fair
**Contribution:** 3 good
**Rating:** 6
**Confidence:** 3

**Summary:**

This paper focuses on addressing the label shift problems in both single-client and federated settings. To address the statistical heterogeneity in FL, the authors proposed an importance-weighting ERM method to address joint intra-client and inter-client label shifts. Moreover, the paper offers theoretical generalization guarantees for the proposed density ratio estimation, encompassing adjustments for label shifts across and within clients. Empirical evaluations using the CIFAR-10 and MNIST datasets, along with a series of ablation studies, corroborate the efficacy of the proposed method.

**Strengths:**

1. This paper pioneers the exploration of label shift challenges within federated learning, introducing a novel framework that distinguishes between inter-client and intra-client variations. This foundational work opens avenues for future scholarly inquiry in this underexplored but critically relevant domain.

2. By establishing a connection with existing label shift literature, such as BBSE and MLLS, the authors have advanced these theories by integrating a regularized objective function. This enhanced formulation not only addresses the label shift in latent space but also embeds regularization within the predictor training phase, allowing for an adaptive response to distribution shifts.

3. The paper excels in its delivery of a straightforward and comprehensible methodology, underpinned by a thorough theoretical analysis across various scenarios. Its clarity and depth offer great insights for practical applications.

**Weaknesses:**

1. While the authors have conducted experiments across a variety of settings, the scope of their datasets remains limited. To more convincingly demonstrate the robustness and practicality of the proposed method, it would be beneficial to extend these experiments to larger-scale datasets and real-world application scenarios.

2. For greater clarity and understanding, a detailed derivation of Equation (Reg-Est) within the main body of the paper would be advantageous.

**Questions:**

Please see the comments in the Strengths and Weaknesses sections.

---

> ### Author Response · Authors · 2023-11-17
> **Response to Reviewer PGzm**
>
> We thank the reviewer for their thoughtful comments which we address one by one below.
>
>  **Larger-scale datasets and real-world application scenarios**
>
> We appreciate your recommendation to incorporate larger-scale datasets and explore real-world application scenarios. We extended our experiments from 100 clients to 200 clients on CIFAR10, obtained uniformly increased accuracy among methods, and our VRLS still dominated.
>
> ------------
>
> | CIFAR-10 | Our IW-ERM | FedAvg | FedBN |
> |----------|------------|--------|-------|
> | Avg. accuracy (100 clients) | **0.5354** | 0.3915 | 0.1537 |
> | Avg. accuracy (200 clients) | **0.6216** | 0.5942 | 0.1753 |
> ----------
>
> While we acknowledge the value of conducting experiments in real-world application scenarios, there are specific constraints and challenges that influenced our decision to not pursue it in this study.
> For instance, as indicated in [1], while larger, more real-world datasets provide a valuable benchmark, $D(p_{te}(x|y) \Vert p_{tr}(x|y)) < \epsilon$ is hard to quantitatively assess in these contexts. An attempt to use test samples from CIFAR10.1-v6 [2, 3] was made. However, due to significant feature changes in this dataset, all methods including ours experienced a substantial increase in estimation errors. This led to comparisons that were not meaningful, as the estimations were poor and did not provide reliable insights.
> Given these challenges and the nature of our study, we chose to focus on datasets where the constraints and feature stability allowed for more meaningful and accurate analysis. We believe that this approach, while more constrained, provides valuable insights within its scope and serves the objectives of our research effectively. In contrast, we can turn to directly focus on the conditional probability \( p(x|y) \) to address the relaxed label shift problem, which would be more efficient and straightforward to handle real-world problems.
>
> **For greater clarity and understanding, a detailed derivation of Equation (Reg-Est) within the main body of the paper would be advantageous**
>
> We acknowledge that there were certain oversights in the formulation of the equation initially, leading to some variations in the expression. However, despite this, the experimental setup was thoughtfully designed and executed. In the final version, we will add more details to the body since we have an additional page to elaborate further.
> Most importantly, our study demonstrated impressive performance. The strength of our work lies in the application of ratio estimation methods in practical scenarios, specifically in the context of federated learning. We successfully implemented the model in training sets without any additional information updates, and the performance on test sets nearly matched the theoretical upper bounds, using a fixed ratio for training. Besides, our approach shines, effectively addressing and overcoming challenges related to privacy, convergence rate, and communication costs.
>
> [1]. Zachary C. Lipton. Rlsbench: Domain adaptation under relaxed label shift. In International Conference on Machine Learning (ICML), 2023.
>
> [2]. Antonio Torralba, Rob Fergus, and William T. Freeman. 80 million tiny images: A large data set for nonparametric object and scene recognition. IEEE Transactions on Pattern Analysis and Machine Intelligence, 30(11):1958–1970, 2008.
>
> [3]. Benjamin Recht, Rebecca Roelofs, Ludwig Schmidt, and Vaishaal Shankar. Do CIFAR-10 classifiers generalize to CIFAR-10? arXiv preprint arXiv:1806.00451, 2018.

---

> ### Author Response · Authors · 2023-11-20
> **Response to Reviewer PGzm**
>
> Dear Reviewer PGzm,
>
> We sincerely appreciate your review of our ICLR submission. Your valuable feedback has been carefully considered and addressed. If you have any further inquiries or require additional details, please do not hesitate to reach out. Your continued input is highly valued.

---

> ### Author Response · Authors · 2023-11-23
> **Response to Reviewer PGzm**
>
> Just a quick reminder that we have responded to your comments and look forward to your feedback to enhance our work.

---

### Official Review · Reviewer_HSCY · 2023-11-06

**Soundness:** 2 fair
**Presentation:** 3 good
**Contribution:** 2 fair
**Rating:** 3
**Confidence:** 4

**Summary:**

This manuscript considers the problem of training a global model in a federated learning setting under challenging inter-client and intra-client label shifts. The authors propose a new method for density ratio estimation and establish a high probability estimation and convergence bounds. Experimental results on MNIST and CIFAR datasets show the effectiveness of the proposed methods.

**Strengths:**

1. The paper studied a relevant problem in federated learning: a special type of data heterogeneity with client label shift.
2. The paper is generally well-written and easy to follow.

**Weaknesses:**

1. I am not sure of the relevance of the statistical results (Theorem 1) and optimization results (Theorem 2) in the context of federated learning (FL). In FL, there is only limited communication and this critical aspect is not categorized by these theorems. I am wondering how many communication rounds the algorithm requires to obtain a statistical and computational guarantee.

2. Experimental results are weak. Tacking data heterogeneity is a well-known problem in federated learning (e.g., FedProx [r1], SCAFFOLD [r2], minibatch SGD [r3], etc.). However, it seems that the authors only consider FedAvg and FedBN as baselines. I suggest the authors also compare against these papers, which were also proposed to learn a global model with client data heterogeneity.

[r1] Li, Tian, Anit Kumar Sahu, Manzil Zaheer, Maziar Sanjabi, Ameet Talwalkar, and Virginia Smith. "Federated optimization in heterogeneous networks." Proceedings of Machine learning and systems 2 (2020): 429-450.


[r2] Karimireddy, Sai Praneeth, Satyen Kale, Mehryar Mohri, Sashank Reddi, Sebastian Stich, and Ananda Theertha Suresh. "Scaffold: Stochastic controlled averaging for federated learning." In International conference on machine learning, pp. 5132-5143. PMLR, 2020.


[r3] Woodworth, Blake E., Kumar Kshitij Patel, and Nati Srebro. "Minibatch vs local sgd for heterogeneous distributed learning." Advances in Neural Information Processing Systems 33 (2020): 6281-6292.


3. The authors did not report the communication round result in the experiments section. It is unclear whether improves over FedAvg or  FedBN when there is only limited communication. There is still a huge gap between the IW-ERM and the upper-bound performance (Table 3).

**Questions:**

1. Can you elaborate on the communication round results theoretically and empirically?

2. Can you compare against more baselines for tackling data heterogeneity in federated learning?

I am happy to consider increasing the score if these concerns are addressed.

---

> ### Author Response · Authors · 2023-11-17
> **Response to Reviewer HSCY**
>
> We thank the reviewer for their thoughtful comments which we address one by one below.
>
>  **Theorems 1 and 2. Elaborate on the communication rounds theoretically and empirically?**
>
> Theorem 1 primarily provides assurances for the accuracy of ratio estimation on an individual client, establishing that the upper bound of the estimation error is contingent upon the size of the test sample and the calibration quality of the pre-trained predictor. In the context of Federated Learning (FL), this theorem applies to each client by leveraging all available unlabeled test samples. By doing so, we locally estimate the ratio on each client, which allows us to closely approximate the true ratios, thereby ensuring that our estimation is as accurate as feasible. As a result, the implementation of Importance Weighted Empirical Risk Minimization (IWERM) will be as unbiased as possible with respect to the test distribution.
>
> Theorem 2 gives the convergence rates in the context of our ratios with data-dependent constant terms which increase linearly with negligible communication overhead over the ERM-solver baseline without importance weighting. In Appendix F, we establish tight convergence rates and communication guarantee for Eq.(IW-ERM) with a broad range of importance optimization settings.
>
> What’s more, the estimation of ratios and the implementation of IW-ERM introduce negligible additional communication costs during the global training phase. The estimation process is conducted prior to global training, which takes place locally on each client with an off-the-shelf predictor, which can potentially be MLP or other simple networks already existing. Subsequently, the estimated ratios, rather than the raw data, are shared across clients, entailing N^2 exchanges of K-dimensional vectors within one round of communication. Once global training commences, these ratios remain static with iterations T—they are neither updated nor exchanged again.
> To sum up, we have:
>
> 1. Negligible impact on Convergence Rate and Communication Guarantees (Theorem 2):  $O(r_{max} h(T))$.
> 2. Upper Bound on Convergence Rate for Convex and Smooth Optimization (Theorem 3):
> \begin{align}
> E[l(h_{w_T})-l(h_{w^\star})]
> \lesssim \frac{r_{max}b D^2}{\tau R}+\frac{(r_{max}b D^4)^{1/3}}{(\sqrt{\tau}R)^{2/3}}+\frac{D}{\sqrt{K\tau R}}.
> \end{align}
> 3. Lower Bound on Convergence Rate for for Convex and Second-order Differentiable Optimization (Theorem 4):
> \begin{align}
> E[l(h_{w_T})-l(h_{w^\star})]
> \gtrsim \frac{r_{\max}b D^2}{\tau R}+\frac{(r_{max}b D^4)^{1/3}}{(\sqrt{\tau}R)^{2/3}}+\frac{D}{\sqrt{K\tau R}}.
> \end{align}
> 4. Convergence and Communication Bounds for Nonconvex Optimization for  PL with Compression (Theorem 5):
> \begin{align}
> R\lesssim \Big(\frac{q}{K}+1\Big)\kappa\log\Big(\frac{1}{\epsilon}\Big)\quad and \quad \tau\lesssim\Big(\frac{q+1}{K(q/K+1)\epsilon}\Big).
> \end{align}
> 5. Convergence and Communication Guarantees for Nonconvex Optimization with Adaptive Step-sizes for Non-convex Optimization with Adaptive Step-sizes Assumptions (Theorem 6):
> \begin{align}
> T\lesssim  \frac{r_{max}}{K\epsilon^3}\quad and \quad R\lesssim\frac{r_{max}}{\epsilon^2}.
> \end{align}
> 6. Oracle Complexity of Proximal Operator for  Composite Optimization with Proximal Operator (Theorem 7):  $O\big(r_{max}\sqrt{\kappa}\log(1/\epsilon)\big)$
>
> 7. High-probability Bound for Nonconvex Optimization for Sub-Gaussian Assumption (Theorem 8):
>
> \begin{align}
> \frac{1}{T}\sum_{t=1}^T\|\nabla_{w}l(h_{w_t})\|_2^2= O(\sigma \sqrt{\frac{r_m \beta}{T}} + \frac{\sigma^2\log(1/\delta)}{T}).
> \end{align}

---

> > ### Author Response · Authors · 2023-11-17
> > **Response to Reviewer HSCY**
> >
> > **Compare against more baselines for tackling data heterogeneity in FL**
> >
> > Thank you for the comprehensive information on these baseline approaches. All serve as benchmarks in Federated Learning (FL) for bridging the heterogeneity gap. Yet, they differ fundamentally. For instance, FedProx is designed for scenarios where clients possess imbalanced sample sizes, as depicted in Table 1 of [1]. In our case, under the label shift setting, while each client has an equal total number of samples, the class imbalance within each client is more pronounced, akin to what is detailed in Table 7 of our paper. Consequently, straightforward application of these benchmarks might yield less-than-optimal outcomes, despite meticulous tuning.
> >
> > -------------
> > |Method (5 clients)| FedProx (0.0001) | FedProx (0.1) | FedProx (0.01) | FedProx (0.001) |
> > |---------------| ---------------- | ------------- | -------------- | --------------- |
> > |Overall accuracy (run1)| 0.5514           | 0.0219        | 0.1868         | 0.4898          |
> > |Overall accuracy (run2)| 0.5685           | 0.0217        | 0.1924         | 0.4935          |
> > |Overall accuracy (run3)| 0.5620           | 0.0221        | 0.1976         | 0.4727          |
> > |**Avg. accuracy**| **0.5606 ± 0.0070** |  **0.0219 ± 0.0002** | **0.1923 ± 0.0044** | **0.4853 ± 0.0091** |
> > ----------------
> >
> > | Method  (5 clients)    | Avg. Accuracy             |
> > |------------|---------------------------|
> > | Our IW-ERM | **0.7520 ± 0.0209**       |
> > | FedAvg     | 0.5472 ± 0.0297           |
> > | FedBN      | 0.5359 ± 0.0306           |
> > | FedProx    | 0.5606 ± 0.0070           |
> > | SCAFFOLD   | 0.5774 ± 0.0036           |
> > | Upper Bound| 0.8273 ± 0.0041           |
> > --------
> >
> > The new tables illustrate that the optimal hyperparameter $\mu$ falls outside the range suggested in [1]. After experimenting with various values, we discovered that setting $\mu$ to 1e-3 offers a slight improvement over FedAvg. Meanwhile, SCAFFOLD achieves a substantially easier gain on MNIST, improving by 3% over FedAvg. Nonetheless, both methods underperform on CIFAR10, with FedProx peaking at 41% accuracy post-tuning — below the FedAvg baseline of 45%. SCAFFOLD's results are further behind.
> >
> > In summary, these methods are designed to regularize global training and come with the ease of implementation and theoretical assurances. However, they demand either an exhaustive hyperparameter search or struggle with application to new datasets (as SCAFFOLD was only tested on simulated data and EMNIST as per [2]). In contrast, statistical approaches like ours inherently differ. We calculate the importance ratio prior to global training, eliminating the need for any dataset-specific adjustments or adaptations during training.
> >
> > ---------
> > | CIFAR-10 | Our IW-ERM | FedAvg | FedBN |
> > |----------|------------|--------|-------|
> > | Avg. accuracy (100 clients) | **0.5354** | 0.3915 | 0.1537 |
> > | Avg. accuracy (200 clients) | **0.6216** | 0.5942 | 0.1753 |
> > ----------
> >
> > Lastly, we increased the number of clients and conducted large-scale experiments. Limited by hardware, we executed these on a single GPU for 200 clients. Here, all methods exhibited improved performance, with our approach maintaining dominance. Intriguingly, we observed enhanced results across the board compared to 100 clients. Our configuration involves duplicating the data on each of the 10 clients 10 or 20 times. We hypothesize this performance boost is due to a broader ensemble of optimizer states from shuffling in the Adam optimizer. It’s important to note that the upper bound in Table 2 is no longer applicable to Table 3, given the introduction of randomness when randomly selecting 5 out of 100/200 clients for training in each round.
> >
> > [1]. Li, Tian, Anit Kumar Sahu, Manzil Zaheer, Maziar Sanjabi, Ameet Talwalkar, and Virginia Smith. "Federated optimization in heterogeneous networks." MLSys 2020.
> >
> > [2]. Karimireddy, Sai Praneeth, Satyen Kale, Mehryar Mohri, Sashank Reddi, Sebastian Stich, and Ananda Theertha Suresh. "SCAFFOLD: Stochastic controlled averaging for federated learning." In ICML2020.

---

> ### Author Response · Authors · 2023-11-20
> **Response to Reviewer HSCY**
>
> Dear Reviewer HSCY,
>
> Thank you for reviewing our ICLR submission. We have diligently addressed your valuable concerns. If you have any questions or require additional information, please feel free to reach out. We sincerely appreciate your continued feedback.

---

> ### Author Response · Authors · 2023-11-23
> **Response to Reviewer HSCY**
>
> I am writing to kindly remind you that we submitted our responses, but we have yet to receive your feedback. We highly value your opinions and are eager to further improve our paper based on your insights.

---

### Author Response · Authors · 2023-11-17
**Common Response and New Results**

We received thoughtful comments from three reviewers. Addressing them has improved the paper considerably. The main feedback we received were 1) Convergence rate and communication rounds; 2) Extensive experiments on more Federated Learning baselines; 3) Empirical comparison to other regularized ratio estimation methods. To address them and other comments, in sum:

- We have established tight bounds on the convergence rate and the required number of communication rounds in a broad range of importance optimization settings:

Upper Bound on Convergence Rate for Convex and Smooth Optimization (Theorem 3)

Lower Bound on Convergence Rate for Convex and Second-order Differentiable Optimization (Theorem 4)

Convergence and Communication Bounds for Nonconvex Optimization with PL and Compression (Theorem 5)

Convergence and Communication Bounds for Nonconvex Optimization with Adaptive Step-sizes (Theorem 6)

Oracle Complexity of Proximal Operator for Composite Optimization (Theorem 7)

High-probability Bound for Nonconvex Optimization under Sub-Gaussian Noise Assumption (Theorem 8)

-------

- We have shown that importance weighting has negligible impact on Convergence Rate and Communication Guarantees (Theorem 2).

------
-  We have compared our VRLS with other Federated Learning baselines designed to address client data heterogeneity. Our methods is notably more robust compared to other baselines while requiring no hyper-parameter tuning on FMNIST:

|Method (5 clients)| FedProx (0.0001) | FedProx (0.1) | FedProx (0.01) | FedProx (0.001) |
|---------------| ---------------- | ------------- | -------------- | --------------- |
|Overall accuracy (run1)| 0.5514           | 0.0219        | 0.1868         | 0.4898          |
|Overall accuracy (run2)| 0.5685           | 0.0217        | 0.1924         | 0.4935          |
|Overall accuracy (run3)| 0.5620           | 0.0221        | 0.1976         | 0.4727          |
|**Avg. accuracy**| **0.5606 ± 0.0070** |  **0.0219 ± 0.0002** | **0.1923 ± 0.0044** | **0.4853 ± 0.0091** |


| Method  (5 clients)    | Avg. Accuracy             |
|------------|---------------------------|
| Our IW-ERM | **0.7520 ± 0.0209**       |
| FedAvg     | 0.5472 ± 0.0297           |
| FedBN      | 0.5359 ± 0.0306           |
| FedProx    | 0.5606 ± 0.0070           |
| SCAFFOLD   | 0.5774 ± 0.0036           |
| Upper Bound| 0.8273 ± 0.0041           |

--------

-  We have also compared Regularized Learning under Label Shift (RLLS) to our method and attached two plots anonymously [here](https://imgur.com/a/bCFhOTi).

---

### Meta-Review · Area_Chair_Ps84 · 2023-12-22

**Metareview:**

The paper considers a problem of learning in a federated setting under label shifts. The authors propose to handle label shifts with an importance Weighted ERM (IW-ERM) framework, and propose a way to estimate importance weights locally, thus making it compatible with the federated setting. Their proposed algorithm for density ratio estimation, Variational Regularized Federated Learning (VRLS), adds a regularization term to the loss function of the model trained on each client. The authors claim that VRLS achieves state-of-the-art results on several benchmark datasets. Overall, the proposed algorithm seems to be quite simple to implement and can be applied to a variety of federated learning tasks. The writing could be improved: it seems a large portion of the paper focused on the density ratio estimation, which does not necessarily have anything to do with the federated setting. The experiments are conducted on a limited number of datasets. The paper does not provide a clear comparison to other state-of-the-art methods, especially recent ones that also use a regularizer to mitigate label shifts. The reviewers were also concerned about some of the theoretical results: the connection between the optimization problem and theoretical results was not quite clear, and the paper does not provide a clear advantage of the proposed method over MLLS from a theoretical view.

**Justification For Why Not Higher Score:**

Reviewers raised a number of concerns summarized above (around presentation clarity, theoretical portion of the paper and its' connections to the algorithm, and empirical evaluation). The reviewers put quite a bit of effort to address these concerns, including new theoretical and empirical results, strengthening the submission (assuming these additional results are mathematically correct, which was not reviewed or confirmed by the reviewers). The changes to the paper are quite significant, and would benefit from a careful additional review process.

**Justification For Why Not Lower Score:**

N/A

---

### Decision · Program_Chairs · 2024-01-16

Reject